# The Product neutrality function defining genetic interactions emerges from mechanistic models of cell growth

Lucas Fuentes Valenzuela[1], Paul Francois[2], Jan M Skotheim[1,3]*

[1]Department of Biology, Stanford University, Stanford, United States; [2]Department of Biochemistry and Molecular Medicine, University of Montreal, Montreal, Canada; [3]Chan Zuckerberg Biohub, San Francisco, United States

## eLife Assessment

The paper addresses the question of gene epistasis and asks what is the correct null model for which we should declare no epistasis. By reanalyzing synthetic gene array datasets regarding single and double-knockout yeast mutants, and considering two theoretical models of cell growth, the authors reach the **valuable** conclusion that the product function is a good null model. While the justification of some assumptions is **incomplete**, the results have the potential to be of value to the field of gene epistasis.

*For correspondence:
skotheim@stanford.edu

Competing interest: The authors declare that no competing interests exist.

**Abstract** Genetic analyses, which examine the phenotypic effects of mutations both individually and in combination, have been fundamental to our understanding of cellular functions. Such analyses rely on a neutrality function that predicts the expected phenotype for double mutants based on the phenotypes of the two individual non-interacting mutations. In this study, we examine fitness, the most fundamental cellular phenotype, through an analysis of the extensive colony growth rate data available for budding yeast. Our results confirm that the Product neutrality function describes the colony growth rate, or fitness, of a double mutant as the product of the fitnesses of the individual single mutants. This Product neutrality function performs better than Additive or Minimum neutrality functions, supporting its continued use in genetic interaction studies. Furthermore, we explore the mechanistic origins of this neutrality function by analyzing two theoretical models of cell growth. We perform a computational genetic analysis to show that in both models, the Product neutrality function naturally emerges due to the interdependence of cellular processes that maximize growth rates. Thus, our findings provide mechanistic insight into how the Product neutrality function arises and affirm its utility in predicting genetic interactions affecting cell growth and proliferation.

## Introduction

Genetic analysis has been one of the primary methods scientists use to understand how a cell works. One way this is done is through the analysis of genetic interactions, in which the phenotypic effects of mutations are analyzed both as single mutations and then together as double mutations in the same cell. Genes are then said to interact if their combination produces phenotypes that are different from what is predicted from a generic model combining two non-interacting mutations (*Phillips, 2008*; *Beltrao et al., 2010*; *Costanzo et al., 2019*). In other words, a genetic interaction is identified when combining multiple mutations yields something unexpected. Yet, what should we expect when combining mutations in such a complex system as a living cell?

The expected phenotype of a double mutant predicted from the two single mutants' phenotypes is defined by the neutrality function. In this way, the neutrality function calculates the expected phenotype of a double-mutant strain carrying two non-interacting mutations (*Beltrao et al., 2010*; *Mani et al., 2008*). If the double-mutant phenotype deviates significantly from that given by the neutrality function for two specific mutations, they are then said to interact. A lot of care, therefore, needs to be taken in selecting the appropriate neutrality function, which depends on the context and phenotype to be examined (*Phillips, 2008*). The neutrality function should be defined such that most mutations are categorized as non-interacting. If the neutrality function were not defined this way, the majority of genes would appear to interact, leaving only a few genes with distinct functions. However, decades of cell biological and structural biological analysis have identified specific functions for many genes and their associated proteins. For example, the components of the ribosome or RNA polymerase have the specific task to form these complexes, and metabolic enzymes catalyze specific biochemical reactions. This implies that a judiciously selected neutrality function should predict most double-mutant fitnesses from the single-mutant fitnesses since any two randomly selected genes should be unlikely to interact. Recent technological advances have enabled the screening of the proliferation of single and double genetic mutants at increasingly larger scales in *E. coli* (*Typas et al., 2008*; *Butland et al., 2008*; *Babu et al., 2014*), fission yeast (*Roguev et al., 2008*; *Dixon et al., 2008*), *C. elegans* (*Lehner et al., 2006*; *Byrne et al., 2007*), and human cells (*Horlbeck et al., 2018*). In budding yeast, Synthetic Genetic Arrays (SGAs) *Tong et al., 2001* have generated the largest and most comprehensive such datasets (*Baryshnikova et al., 2010b*; *Costanzo et al., 2010*; *Costanzo et al., 2016*).

The most fundamental phenotype of a cell is its fitness, namely how quickly it grows and proliferates in a given environment. Here, we focus on this property and define fitness as the relative exponential growth rate with respect to that of a wild-type cell. For yeast proliferation, single-mutant fitnesses are usually assumed to combine according to a Product neutrality function, namely that the fitness of the double mutant is the product of the fitnesses of the single mutants (*Mani et al., 2008*). This neutrality function was shown to better predict double-mutant fitnesses than an Additive neutrality function, where the differences between wild-type and mutant fitnesses were simply added together, and a Minimum neutrality function, where the double-mutant fitness was taken as the lowest fitness of the two single-mutant strains. However, this analysis was based on older, less extensive data, which raises the question of whether this neutrality function remains accurate when the large amount of more recently collected yeast data is also considered. And, if so, then why does the Product neutrality function accurately describe double-mutant fitnesses? In other words, what are the properties of the underlying system controlling cell growth and proliferation that result in a Product neutrality function for mutant fitnesses?

In this paper, we conduct an in-depth analysis of recent yeast double-mutant datasets and show that they support the Product neutrality function. Moreover, we analyze two theoretical models of growth of increasing complexity (*Scott et al., 2010*; *Weiße et al., 2015*) and find that the Product neutrality function emerges naturally from both growth models, albeit with small deviations specific to each. Taken together, our work supports the use of the Product neutrality function to model genetic interactions in the regulation of cell growth and gives mechanistic insight into its origin.

## Results

### High-throughput gene perturbation experiments in budding yeast support a Product neutrality function for double-mutant fitness

To test the general validity of neutrality functions for mutations affecting cell proliferation, we sought to examine the most extensive such dataset. In the SGA dataset, over 20 million single- and double-mutant budding yeast strains were generated. Then, the growth rates of their colonies were measured in SGAs (*Tong et al., 2001*; *Baryshnikova et al., 2010b*; *Costanzo et al., 2010*; *Costanzo et al., 2016*). Each mutant's fitness was then defined as this measured growth rate normalized by that of the wild-type strain, enabling a consistent comparison across thousands of genotypes. Next, we use SGA datasets of growth of single- and double-mutant cells for pairs of mutations to test specific neutrality functions.

Here, we followed (*Mani et al., 2008*) and began our examination using the Product, Additive, and Minimum neutrality functions (see *Figure 1A*). The Product neutrality function predicts that the

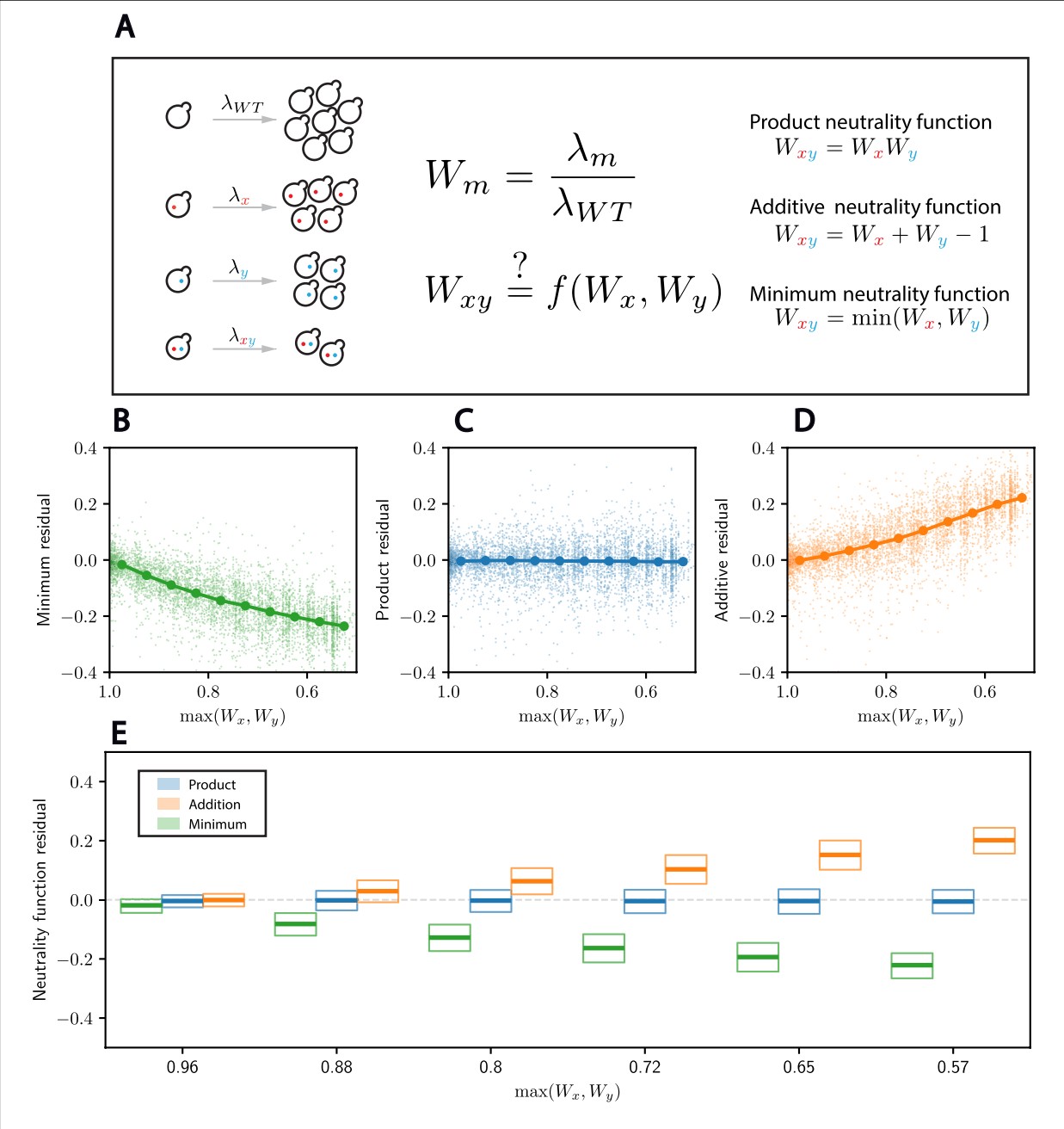

**Figure 1.** High-throughput gene deletion experiments in budding yeast support a Product neutrality function for double-mutant fitness. (**A**) Budding yeast mutant fitness is defined as the colony growth rate relative to that of wild-type cells. Schematic illustration of epistasis in growth rate and of different laws proposed in the literature. *λ* denotes the growth rate, and *W* the fitness. (**B–D**) For each double mutant, we plot the residual of the fitness predicted from the indicated model against the fitness of the fittest of the two separate single mutants (maximum single-mutant fitness). Dots indicate the median for 10 equally spaced bins between 0.5 and 1. (**E**) Box plots for the distributions of the residuals for the three neutrality functions as a function of the maximum single-mutant fitness. A thick line denotes the median, and boxes denote the 25th and 75th percentiles of the distributions. The data plotted here represents a subset of the entire Synthetic Genetic Array (SGA) dataset, corresponding to the Deletion Mutant Array (DMA) at 30°C.*Appendix 1—figures 1 and 2* report results for the other subdatasets.

fitness of a double mutant is the product of the fitnesses of the two corresponding single mutants, while the Additive neutrality function proposes that the difference between the double-mutant and wild-type fitnesses is the sum of the differences between the two mutant and wild-type fitnesses. The Minimum neutrality function proposes that the fitness of the double mutant is equal to the fitness of the least fit single mutant. Effectively, these neutrality functions express different forms of modularity

or independence between cellular processes. The Product model suggests that a mutation's effect depends on the fitness of the background strain without that mutation, while a mutation's effect is independent of the fitness of the background strain in an Additive model. The Minimum model suggests that there is some rate-limiting process whose slow time scale dominates the determination of cell growth so that more minor mutations affecting other processes have no additional effect.

To compare the different neutrality functions with double-mutant fitnesses, we first perform some minor pre-processing of the SGA data (see Methods for details). We note that these data are from combinations of gene deletions, temperature-sensitive alleles, and hypomorphic mutants (see Methods). Moreover, cells were growing quickly on the relatively rich synthetic complete media containing glucose (*Baryshnikova et al., 2010a*). For these reasons, increasing the growth rate is difficult, and we are in the regime where mutations generally decrease fitness. This contrasts with evolution experiments, where fitness increases very slowly through the gradual accumulation of mutations, which likely exhibit different neutrality functions from those we consider here (*Phillips, 2008*; *Bakerlee et al., 2022*; *Johnson et al., 2023*). Consistent with previous work (*Mani et al., 2008*), we see that the Product neutrality function better predicts double-mutant fitnesses as a function of the single-mutant fitnesses over a broad range of fitness defects (*Figure 1*, *Appendix 1—figure 1*). Indeed, the median residual for the Product neutrality function remains very close to zero even for highly deleterious mutations, while it significantly deviates for the other two. For instance, for a maximum single-mutant fitness of 72%, the median residual is −0.8% for the Product neutrality function, while they are −16.8% and 9.8% for the Minimum and Additive neutrality functions, respectively. For smaller values of the maximum single-mutant fitness, the median residual remains virtually unchanged for the Product neutrality function, while it deviates even further from zero for the other

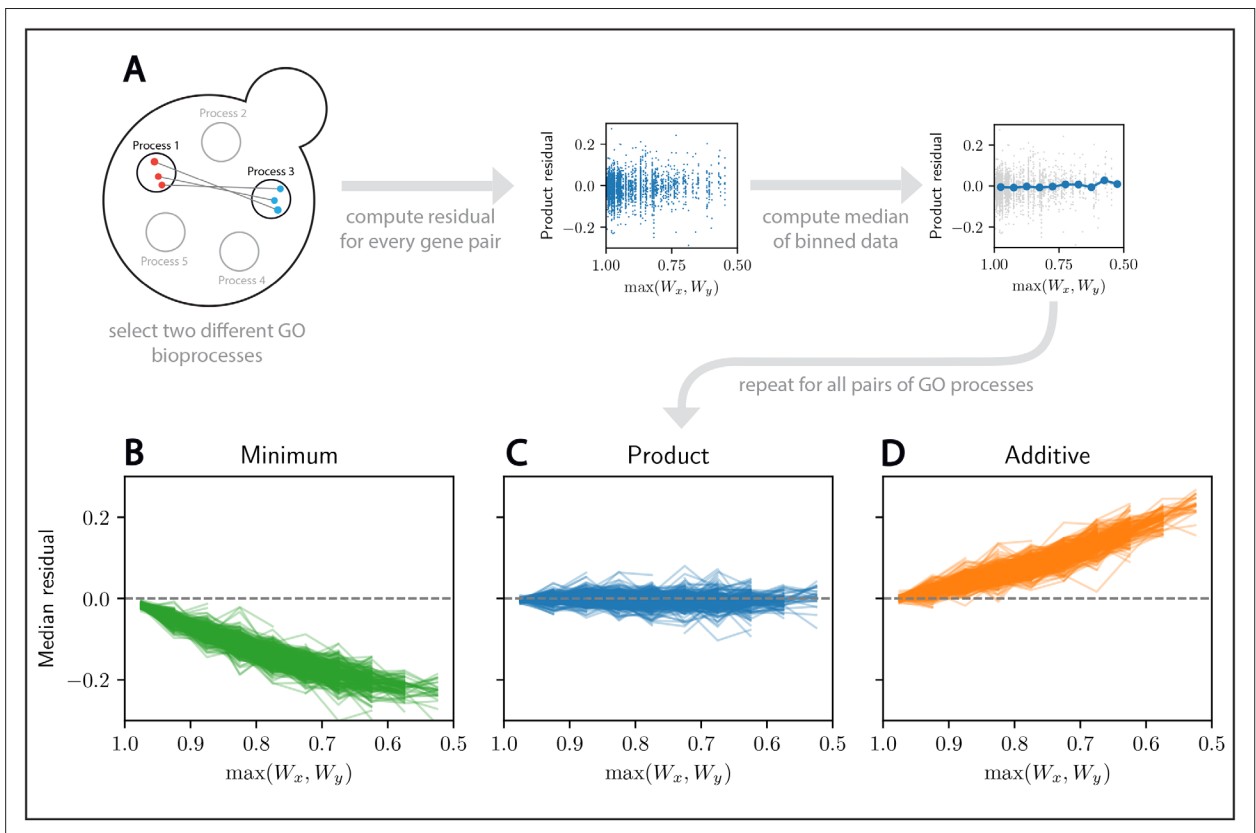

**Figure 2.** The Product neutrality function describes interactions between genes associated with two distinct biological processes. (**A**) Schematic illustration of the analysis process. We first select two different Gene Ontology (GO) biological processes and extract the double mutants in the Synthetic Genetic Array (SGA) dataset associated with them. Then, we compute the median residual for each pair of biological processes and each neutrality function. (**B–D**) Median residual for the Minimum, Product, and Additive neutrality functions as a function of the maximum single-mutant fitness. Each line denotes mutations to a different pair of distinct GO biological processes. The majority of biological process pairs closely follow the Product model.

two. However, we note that significant variation around the median residual remains. At a maximum single-mutant fitness of 72%, the interquartile range lies between 8% and 10% depending on the neutrality function considered.

This observation that the Product neutrality function best describes double mutant fitnesses holds for mutations to essential and nonessential genes, and across different temperature conditions (see Methods and *Appendix 1—figures 1 and 2*). The Minimum neutrality function generally predicts fitnesses that are too high, while the Additive neutrality function generally predicts fitnesses that are too low.

## The Product neutrality function describes interactions between genes associated with two distinct biological processes

While the Product neutrality function predicts double-mutant fitnesses better than the other ones we considered, there remains significant variation (residuals) in the data. This suggests that mutations affecting different functional parts of the cell might be following different neutrality functions. To determine whether this is the case, we analyze the distribution of epistasis residuals for pairs of distinct biological processes and their associated genes. We use the Gene Ontology (GO) dataset (*Ashburner et al., 2000*, *Aleksander et al., 2023*) and, for each biological process, extract the genes in the SGA dataset that are associated with this process (see Methods for details). In particular, we define inter-process gene pairs as pairs of gene perturbations in two distinct biological processes identified using GO annotations (*Figure 2A*). Similarly, intra-process pairs are defined as pairs of gene perturbations in the same GO-annotated biological process (*Appendix 1—figure 3A*). Then, for each pair of GO-defined processes, we compute the residuals for the neutrality functions for all pairs of mutations where one mutation is associated with one process and the second mutation with the other. For each neutrality function and pair of GO processes, we extract the median residual as a function of the largest single-mutant fitness defect (a proxy for mutation severity). This shows that, while imperfect, the Product neutrality function is a good description of typical interactions, while the Additive and Minimum neutrality functions have large, systematic residuals (*Figure 2B–D*). In general, this result is expected and consistent with the use of this type of genetic analysis to define mutations in genes from different biological processes as not interacting. Moreover, we find that this is not only generally true, but also true for each specific pair of processes that we consider. In other words, we do not find evidence that there are particular pairs of processes whose mutations significantly deviate from the Product neutrality function or more closely follow an Additive or Minimum neutrality function.

While mutations associated with different biological processes have fitnesses generally predicted by the Product neutrality function, since they generally do not interact, this may not be the case for mutations associated with the same process. For example, if two mutations break the same protein complex, one would not expect any additional drop in fitness for the double mutant. Consistent with this notion, for gene pairs in the same biological process, we observe more deviations as well as significantly larger residuals (*Appendix 1—figure 3B–D*). The quantitative comparison of the two types of interactions reveals that large residuals (both positive and negative) are significantly more likely for two mutations categorized as being in the same GO process (*Appendix 1—figure 3E, F*).

We note that the SGA dataset has already been used to assign a biological process to each gene (*Costanzo et al., 2016*). For each mutation, a vector of residuals for the Product neutrality function with all other mutations was generated. Then, a Pearson correlation coefficient was calculated for each pair of these vectors. The reasoning was that mutations affecting the same biological process should have similar genetic interaction profiles, which was found to be the case. This then allowed the clustering of groups of correlated mutations, which were named using prior knowledge of many genes in each cluster. That this analysis generally works, that is, the gene clusters have discernible biological meaning, can be viewed as further support of the Product neutrality function.

## A bacterial growth model partially supports the Product neutrality function

Having verified empirically that the Product neutrality function is supported by the latest data for cell proliferation, we now turn our attention to its origins. Addressing this question requires some mechanistic model of biosynthesis. However, most mechanistic models of growth apply directly to single cells in rich nutrient conditions, which may not directly apply to the SGA measurements of

colony expansion rates. In particular, colony growth has been shown to follow a biphasic pattern (*Meunier and Choder, 1999*). A first exponential phase is followed by a slower linear phase as the colony expands. Previous modeling and empirical work indicates that this second linear expansion rate reflects the underlying exponential growth of cells in the periphery of the colony (*Pirt, 1967*; *Gray and Kirwan, 1974*; *Baryshnikova et al., 2010a*; *Gandhi et al., 2016*; *Zackrisson et al., 2016*; *Miller et al., 2022*). More precisely, mathematical models show the linear colony-size expansion rate is directly proportional to the square root of the exponential growth rate under non-limiting conditions. Intuitively, this relationship arises because colony growth is dominated by the expansion of the population of cells in an annulus at the colony border that are exposed to rich nutrient conditions. These cells expand at a rate similar to the exponential rate of cells growing in a rich nutrient liquid culture. In contrast, the cells in the interior of the colony experience poor nutrient conditions, grow very slowly, and do not contribute to colony growth.

This intimate relationship between both proliferation rates allows us to explore the origin of the Product neutrality function in mechanistic models of cell growth. Indeed, if colony-based fitnesses follow a Product model, then

$$W_{xy}^c \sim W_x^c W_y^c \Leftrightarrow \frac{\lambda_{xy}^c}{\lambda_{WT}^c} \sim \frac{\lambda_x^c \lambda_y^c}{(\lambda_{WT}^c)^2},$$

where the superscript $c$ indicates colony-based values for the fitness $W$ and the growth rate $\lambda$. Taking into account the relationship between single-cell exponential growth rates and colony growth rates, we can write

$$\lambda^c \propto \sqrt{\lambda^l},$$

where the superscript $l$ denotes liquid cultures. Combining these expressions, we obtain

$$\frac{\sqrt{\lambda_{xy}^l}}{\sqrt{\lambda_{WT}^l}} \sim \frac{\sqrt{\lambda_x^l}\sqrt{\lambda_y^l}}{\sqrt{\lambda_{WT}^l}^2} \Rightarrow W_{xy}^l \sim W_x^l W_y^l.$$

In other words, from the perspective of the Product neutrality function, fitnesses based on colony expansion rates are equivalent to fitnesses based on single-cell exponential growth rates. The prevalence of the Product neutrality model—both in the SGA data and in previous studies on datasets from liquid cultures (*Mani et al., 2008*; *Jasnos and Korona, 2007*; *Onge et al., 2007*)—encourages the exploration of its origin in mechanistic models of cell growth.

While models of entire cells do exist, these models are complex and computationally intensive (*Onge et al., 2007*; *Karr et al., 2012*). This makes probing and extracting explanatory information from these models difficult. We therefore sought to analyze simpler, more tractable, lower-dimensional models of cell growth. Coarse-grained models offer an appealing alternative for probing the fundamental principles of metabolism and growth (*Scott et al., 2010*; *Weiße et al., 2015*; *Roy et al., 2021*; *Balakrishnan et al., 2022*; *Chure and Cremer, 2023*; *Calabrese et al., 2023*). Rather than representing as many reactions as possible, they provide an integrated representation of generic processes in the cell. Their simplicity and low dimensionality make them easy to compare with empirical measurements and to examine for potential explanatory relationships.

The reduced, tractable model of cell growth that we will consider first was developed for *E. coli* (*Scott et al., 2010*; *Scott and Hwa, 2011*; *Figure 3A*). While this model was developed for *E. coli* bacteria and validated using data from this organism, there is nothing specific to prokaryotes in the model. Experimental measurements in other organisms suggest that the observations leading to this model, including that the cellular ribosome fraction increases with growth rate, are in fact generic and also seen in the yeast *S. cerevisiae* (*Metzl-Raz et al., 2017*; *Elsemman et al., 2022*; *Xia et al., 2022*). In its simplest form, the model defines growth as resulting from two sets of processes, metabolic and translational, that interact in a linear pathway. The metabolic sector provides precursors that are then assembled into proteins by the translational sector, and the flux through each sector is determined by the amount of proteins in that sector. For optimal growth, in which no proteins are wasted, the flux through the metabolic sector is equal to the flux through the translational sector so that

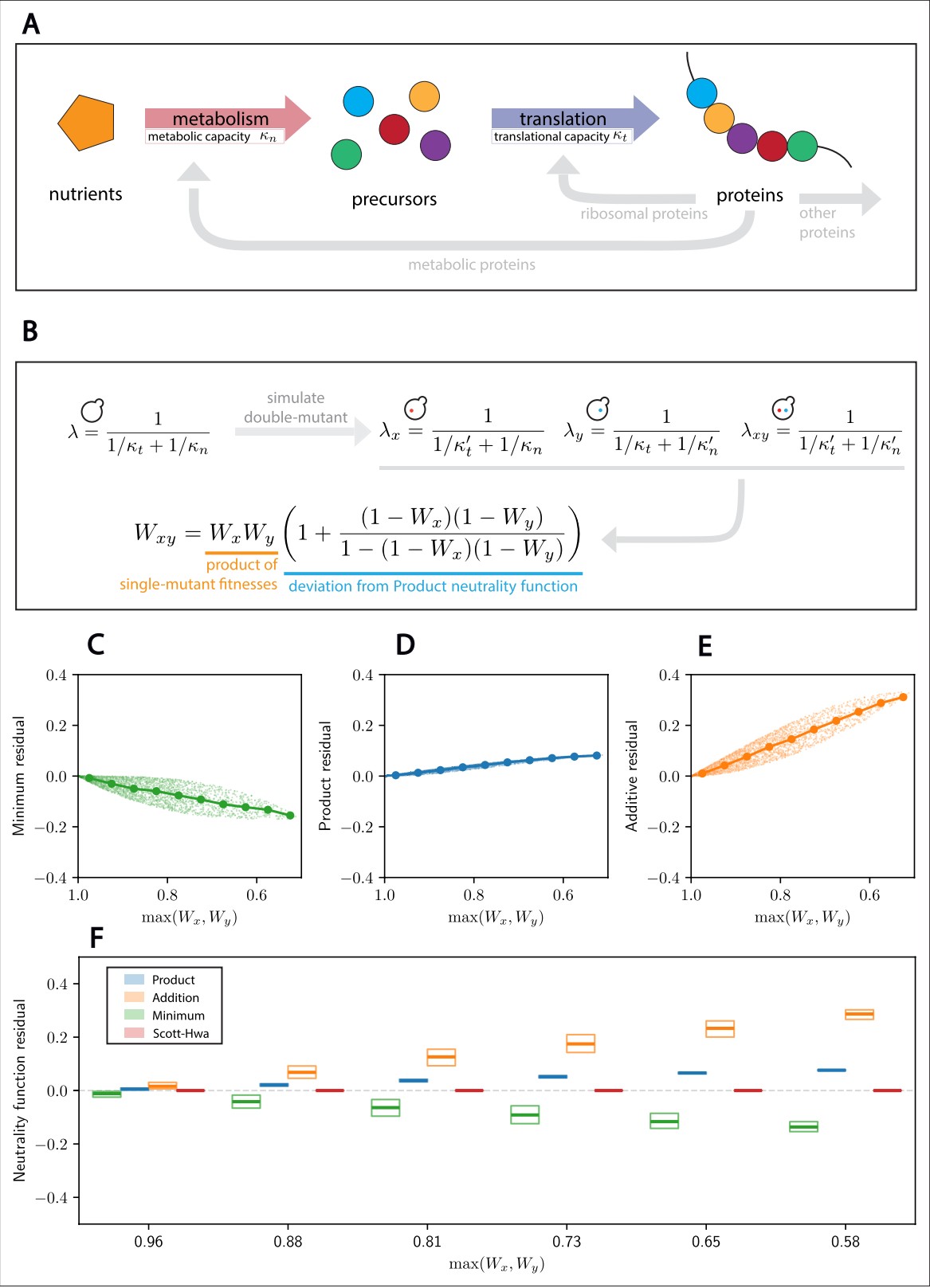

**Figure 3.** A bacterial growth model partially supports the Product neutrality function. (**A**) Schematic of the bacterial growth model by Scott and Hwa. Growth rate is defined by the translation flux, which is itself equal to the metabolic flux. The cell partitions its proteome so as to maximize growth rate. (**B**) Mutations are modeled such that they affect either of the parameters, separately. Values of $\kappa_t$ and $\kappa_n$ in the mutant are indicated with primes and are sampled from a uniform distribution from 0 to their value in wild-type cells. indicates the corresponding growth rate. The analytical expression of

*Figure 3 continued on next page*

*Figure 3 continued*

the double-mutant fitness consists of the Product model with a perturbation. (**C–E**) For each sampled double mutant, we plot the residual of the fitness predicted from the indicated model against the fitness of the fittest of the two separate single mutants (maximum single-mutant fitness). Dots indicate the median for 10 equally spaced bins between 0.5 and 1. (**F**) Box plots for the distributions of the residuals for the three models and the model in (C) as a function of the maximum single-mutant fitness. A thick line denotes the median, and boxes denote the upper and lower quartiles of the data. The analytical model in (B), named Scott–Hwa and shown in red, is exact.

$$\lambda = \kappa_t \phi_t = \kappa_n \phi_n,$$

where the flux through translation and metabolic sectors is characterized by the parameters $\kappa_t$ and $\kappa_n$ multiplying the fraction of the proteome devoted to ribosomes and metabolic proteins, $\phi_t$ and $\phi_n$, respectively. These fluxes directly determine the growth rate of the cell, $\lambda$. The total proteome is fixed and is partitioned into metabolic, translational, and 'other' sectors of the cell so that

$$1 = \phi_t + \phi_n + \phi_o,$$

where $\phi_o$ is the fraction of the cell devoted to other housekeeping functions. This set of algebraic equations can be solved for the optimal growth rate

$$\lambda = (1 - \phi_o) \frac{1}{1/\kappa_t + 1/\kappa_n}.$$

We can then model a mutation as a perturbation to these parameters that decreases the growth rate since these are the types of mutations that dominate the budding yeast data (*Figure 3B*). Single mutants have either $\kappa_t$ or $\kappa_n$ perturbed, while double mutants have both parameters perturbed. Given that we are concerned with two non-interacting mutations, we do not consider the cases where both mutations affect the same parameters. Indeed, one expects two mutations affecting the same parameter to interact. These combinations are therefore inappropriate to study the emerging neutrality functions from growth models, and we do not consider them in this paper.

We can also consider mutations to $\phi_o$, which could be associated with deleting a gene encoding a protein that is not required for the given growth condition. This would serve to increase the cell growth rate because now a larger fraction of the proteome could be devoted to metabolism and translation. In this case, a mutation to $\phi_o$ and another to either $\kappa_t$ or $\kappa_n$ would combine exactly multiplicatively so that

$$W_{xy} = W_x W_y.$$

However, in the generally rich media conditions the SGA experiments were done, there is no evidence that any gene deletion causes an increase in cell growth rate so we do not consider this type of mutation further. We therefore ignore the multiplicative factor $(1 - \phi_o)$ and analyze the following expression for growth rate

$$\lambda = \frac{1}{1/\kappa_t + 1/\kappa_n}.$$

We can then analytically derive a closed-form solution for the double-mutant fitness as a function of the single-mutant fitnesses (*Figure 3C*; see SI for details). Under this model, which we call Scott–Hwa in reference to the authors of the initial work, we observe that the Product neutrality function fits the mutational analysis of the Scott–Hwa model better than the Additive or Minimum neutrality functions (*Figure 3D-G*).

We understand the better performance of the Product neutrality function to arise from a type of feedback regulation that ensures that the flux through all sectors is equal. This effectively makes the mutations to the metabolic and translational sector interdependent, despite their a priori independent functions. In this model, these sectors are coupled because the cell is assumed to have a feedback process to optimize growth rate under any perturbation to its parameters. For instance, in the case of a mutation that decreases the metabolic capacity $\kappa_n$, this feedback drives an increase in the fraction of metabolic proteins at the expense of translational proteins such that the growth rate is maximized under those new parameters. If this feedback were absent, then the growth rates of the double mutant would be significantly lower. In that case, the double-mutant fitness actually follows a

Minimum neutrality function (see Appendix 1 and *Appendix 1—figure 4*). Finally, we also note that the Product neutrality function does not accurately predict model fitnesses well for beneficial mutations—that is, mutations that increase growth rate (see *Appendix 1—figure 5*). This is because the deviations from the Product neutrality function in the mathematical derivation for the double-mutant fitness in *Figure 3* can diverge for beneficial mutations (e.g. $W_x = W_y = 2$). When the Scott–Hwa model's growth-optimizing feedback operates in the context of beneficial mutations, as one process is made more efficient, proteomic resources are allocated to accelerate other processes in the cell. In this way, improving the efficiency of one process will indirectly benefit other processes, leading to compound effects such that double-mutant fitnesses are higher than any of the three models predicts for beneficial mutations.

We note that in this analysis, we do not aim to replicate the statistics of mutations in the SGA dataset, where mutations to either sector could be statistically rarer or more frequent than the other. Instead, we here aim to analyze how mutations to independent parameters governing cell growth combine considering the simplest model with two sectors and their corresponding parameters.

## The Product neutrality function accurately predicts fitness for many pairs of parameters in a more complex cell growth model

While the Scott–Hwa model has proven successful for predicting many aspects of bacterial growth, it remains very simple. Therefore, we sought to explore a more complex model that explicitly incorporates more aspects of biosynthesis. Here, we consider the model of *Weiße et al., 2015*, which incorporates nutrient intake, transcription, competitive binding between mRNAs and ribosomes, and translation, all of which are mediated by associated enzymes and a limiting cellular 'energy' (*Figure 4A*). The Weiße model decomposes cell growth into multiple steps (see supporting material for a full model description). External nutrients are first imported into the cell and then metabolized into a cellular 'energy'. Both of these steps are catalyzed by associated transport and metabolic enzymes according to Michaelis–Menten kinetics. Transcription and translation are then activated by this generated 'energy', also via Michaelis–Menten kinetics. In particular, the model incorporates different transcription rates for ribosomal and non-ribosomal mRNAs. Different mRNAs then compete for free ribosomes to form a ribosome–mRNA complex. This mRNA competition is modeled using mass action kinetics with specified binding and unbinding rates. Four types of proteins are explicitly modeled as a product of translation: transport proteins, metabolic enzymes, ribosomal proteins, and so-called q-proteins which support housekeeping functions much like the 'other' proteins in the Scott–Hwa model.

To perform a mutational analysis of the Weiße model, we first identified the parameters in the model that can reasonably be expected to change through a gene perturbation (see supporting material for details). For instance, we assume that some parameters, such as the maximum nutrient import rate, could be impacted by a gene perturbation, while other parameters, such as the average gene length in the genome, could not. This led to the identification of 9 easily interpreted parameters whose mutation could negatively impact the cell growth rate and correspond to 28 parameter pairs associated with different biological processes. We then performed a similar analysis as we did for the Scott–Hwa model. Namely, we constructed mutants for each pair of parameters by rescaling the values of the original parameters by a number randomly sampled between 0 and 1 (see Methods for details). We then analyzed the statistics of the epistasis coefficients for the neutrality functions that we considered so far (*Figure 4B*).

Our mutational analysis of the Weiße model revealed several striking observations. First, the Product neutrality function is generally better than the Additive or Minimal neutrality functions at describing the mutational results. However, we observe a range of responses and can identify two key subpopulations of parameter pairs. The subset of parameter pairs involving protein translation follows the Scott–Hwa model very closely (*Figure 4D*). This is not entirely surprising, as the Weiße model is an extension of the Scott–Hwa model and incorporates a similar global feedback optimizing cell growth and a competition for resources. On the other hand, another subset of parameter pairs follows the Product neutrality function even more closely. These parameter pairs involve mutations to the other sectors, including the transport, metabolism, and transcription sectors (*Figure 4C*). This raises the question of why some parameter pairs more closely follow the Product neutrality function than others.

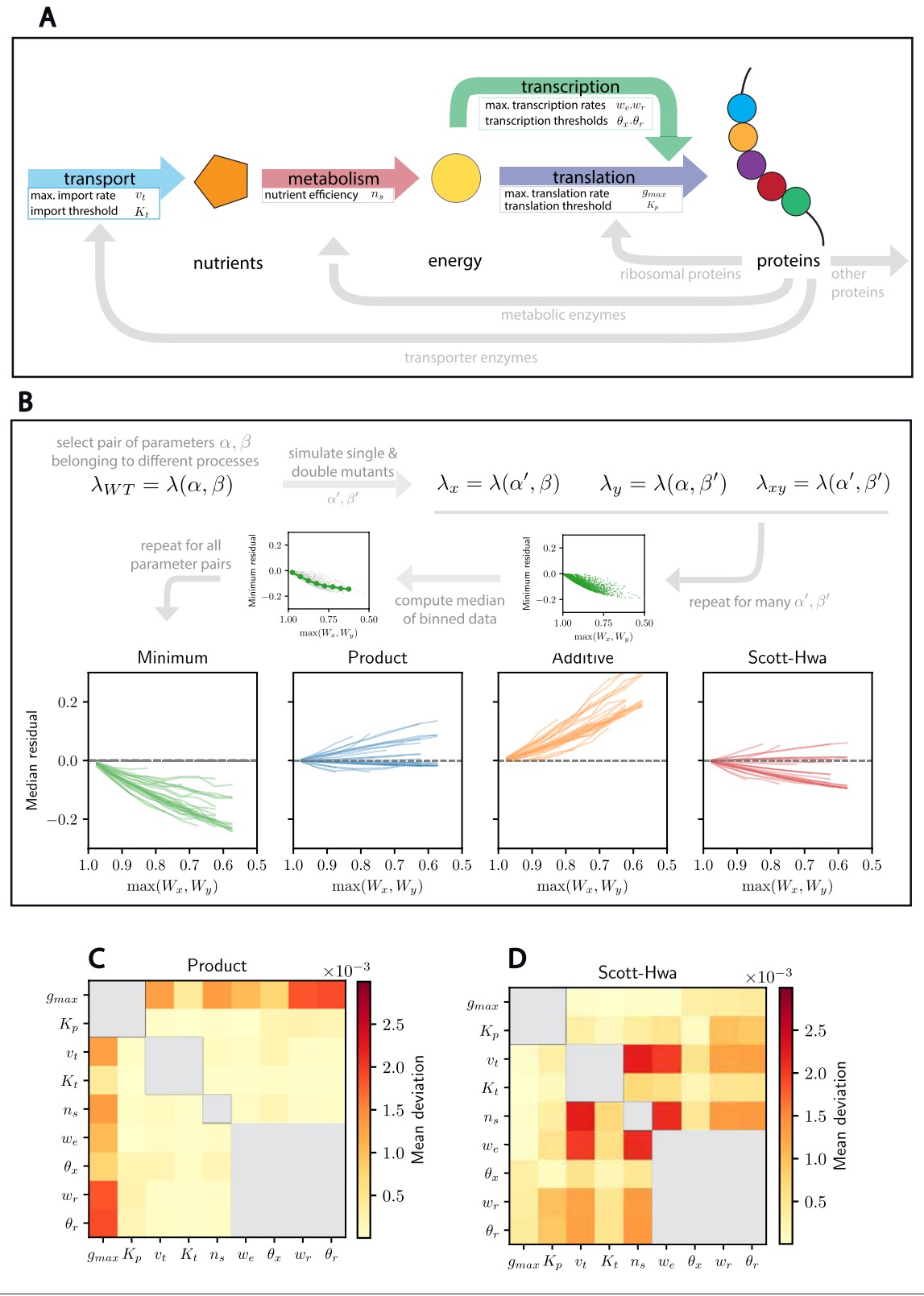

**Figure 4.** The Product neutrality function accurately predicts fitness for many pairs of parameters in a more complex cell growth model. (**A**) Schematic of the growth model from *Weiße et al., 2015*. This model includes nutrient intake, metabolism, transcription, and translation. (**B**) Schematic of the mutational analysis. For each pair of parameters α and β, mutations are modeled such that they affect either of the parameters, separately. Then, the median residual is computed for each neutrality function and they are subsequently reported for every pair of parameters considered. For each

*Figure 4 continued on next page*

*Figure 4 continued*

parameter pair, we report the mean deviation of the simulated double mutants from (**C**) the Product neutrality function and (**D**) the analytical expression of the double-mutant fitness under the Scott–Hwa model. Only parameter pairs corresponding to two different biological processes are considered. Those corresponding to the same process are grayed out. Parameter pairs involving translation ($g_{max}$, $K_p$) are the ones described best by the Scott–Hwa model, while the others are better described by the Product neutrality function.

## Nonlinear kinetics drive deviations from the Product neutrality function in the Weiße model for cell growth

To address the question as to what drives deviations from the Product neutrality function in our genetic analysis of the Weiße model, we took an analytical approach. We examined the dependence of the growth rate on the parameter pairs exhibiting small deviations from the Product neutrality function. To do this, we first extracted a closed form expression that models the growth rate $\lambda$ and its dependence on two mutated parameters $\alpha$ and $\beta$ (**Figure 5A**; see supporting material). While only approximate, this derivation represents the data appropriately for members of this subset of parameters (see *Appendix 1—figure 7*). There are two striking features in this derivation. First, the growth rate $\lambda$ has an explicit dependence on the two parameters $\alpha$ and $\beta$, when $\alpha$ and $\beta$ are selected from the subset of parameters governing metabolism and transport sectors. Second, the amplitude of deviation from the Product neutrality function is governed by an additional parameter, $\gamma$, which is the inverse of the Michaelis–Menten constant giving the transcription rate as a function of the cellular 'energy'. Thus, when $\gamma$ is small, transcription is a less efficient process that is then linearly related to the cellular 'energy' available. When $\gamma$ is larger, transcription is saturated and performed at a rate unrelated to the available 'energy'. As we decrease $\gamma$, we observe that the Product neutrality function is a better and better approximation (**Figure 5C**). Importantly, the intuition provided by the analytical approximation extends to multiple pairs of parameters (see Appendix 1 and *Appendix 1—figure 8*). Taken together, our analysis of the Weiße model shows how the Product neutrality function naturally arises for many different parameter pairs and how deviations from it can be driven by nonlinear effects, such as those that can emerge from Michaelis–Menten kinetics.

## Discussion

Cell growth and proliferation are fundamental to cell biology and have been subject to extensive genetic analysis aiming to understand the underlying regulatory network. Such genetic analysis often aims to identify interactions between mutations through the combination of individual mutations in a double-mutant cell. If the double-mutant proliferates at an unexpected rate, the mutated genes are considered to interact. This, of course, raises the question as to what is the expected rate of proliferation for a cell containing both mutations given the proliferation rate of a cell containing only one of the individual mutations. By analyzing a high-throughput dataset of interactions of gene perturbations in budding yeast (*Costanzo et al., 2010*; *Costanzo et al., 2016*), we found that single-mutant fitnesses tend to combine multiplicatively, consistent with earlier work (*Mani et al., 2008*).

After establishing that the fitness of a double mutant is expected to be approximately the product of the fitness of the individual mutants, namely, the Product neutrality function, we sought to determine if this was also a feature of models of cell growth. If so, then what underlying mechanisms present in these models give rise to this Product neutrality function? Our analysis complements previous, more abstract theoretical attempts at understanding the origin of the Product neutrality function that are not based on any specific model of cell growth (*Chiu et al., 2012*). Indeed, we found that the Product neutrality function best fits the simulated double-mutant fitnesses despite deviations that depend on the specific parameter pairs and the particular model considered.

That the Product neutrality function fits the budding yeast data and cell growth models better than the Additive and Minimum neutrality functions has important implications for the underlying network controlling cell growth and proliferation. On the one hand, the Minimum neutrality function implies that the double-mutant fitness is set by the most deleterious mutation so that the process this gene is involved in becomes rate limiting for cell growth. Clearly, cell growth does not operate this way, likely because the underlying processes are interconnected. Mutations impairing protein translation impact synthesis of all the proteins in the cell so that other processes, like transcription or surface transport, are also affected. As the cell readjusts its machinery to ensure optimal use of resources,

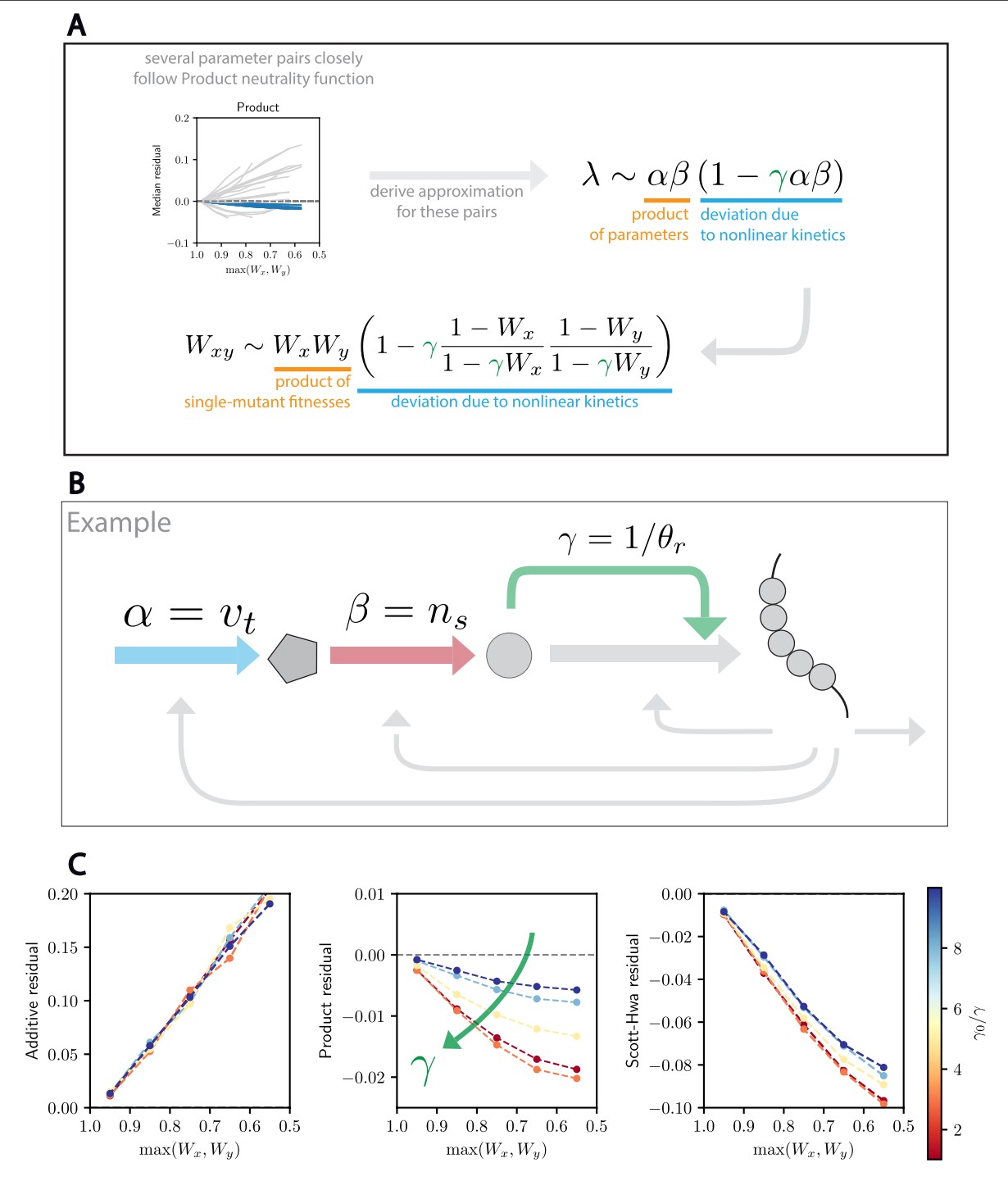

**Figure 5.** Nonlinear kinetics drive deviations from the Product neutrality function in the Weiße model. (**A**) A subset of parameter pairs we analyzed follows the Product model very closely. We derived an analytical approximation of the growth rate and the double-mutant fitness for these pairs and found that the deviation from the product law is governed by nonlinear kinetics. (**B**) In the case of the parameter pair $(v_t, n_s)$, we show that the deviation from the Product model is driven by the Michaelis–Menten constant $\theta_x$ associated with transcription (see Supporting Information). (**C**) Tuning the value of $\gamma$ impacts how good of an approximation the Product neutrality function is for this and other parameter pairs (see text). This analysis validates the analytical approximation and highlights how nonlinear kinetics, in this case Michaelis–Menten kinetics, can drive deviations from the Product neutrality function.

interconnected processes are impacted through a redistribution of cellular resources. On the other hand, the Additive neutrality function implies that a mutation has the same absolute effect on the proliferation rate regardless of the presence of another mutation. This is also clearly not the case as the Additive neutrality function consistently predicts fitnesses below those observed in the data. In the models, this is due in part to growth-supporting feedback that reapportions the proteome. In reality, this may reflect the presence of the general stress response which supports cells in response to genetic or environmental perturbations limiting their growth rate (*Gasch et al., 2000*). In this way, the Product neutrality function is a reasonable intermediate model between Minimum and Additive that incorporates—albeit approximately—effects such as growth-optimizing feedback and is consistent with the phenomenon of *diminishing returns* or *increasing cost* epistasis (*Reddy and Desai, 2021*). Moreover, our theoretical analysis gives insight into the mechanistic underpinning of the Product neutrality function. In our analyses, a product of the single-mutant growth rates naturally emerges in the analytical treatment of both theoretical models that we consider, albeit with deviation terms that depend on the specific model and simplifying assumptions.

Taken together, our work here constitutes a first step toward understanding the structure of interactions inherent in cell growth models. While we focused on coarse-grained models for their simplicity and mechanistic interpretability, they might be too simple to effectively model large double-mutant datasets and the resulting double-mutant fitness distributions. For instance, it is not possible to differentiate between multiple types of growth rate perturbations impacting the same sector, as they would all be modeled through a limited number of parameters (*Metzl-Raz et al., 2017*). We therefore expect the combination of high-throughput genetic data with the analysis of larger-scale models, for instance based on Flux Balance Analysis, Metabolic Control Analysis, or whole-cell modeling, to lead to important complementary insights regarding the regulation of cell growth and proliferation (*Karr et al., 2012*; *Oftadeh et al., 2021*; *Segrè et al., 2005*; *He et al., 2010*; *Orth et al., 2010*; *Kacser and Burns, 1973*; *Szathmáry, 1993*; *Dykhuizen et al., 1987*; *de Vienne et al., 2023*; *Kryazhimskiy, 2021*) We also believe that theoretical exploration of fitness landscapes will shed light on the underlying structure of growth and metabolism networks (*Reddy and Desai, 2021*; *Guo et al., 2019*; *Boffi et al., 2023*). In addition to larger-scale models, we see the refinement of the measurement of cell growth rates as a path forward to a better understanding of its regulation (*MacLean, 2010*). While we showed here that the Product neutrality function fits the data well for deleterious mutations, we anticipate that there are significant and meaningful deviations that are currently obscured by experimental noise. Similarly, large-scale measurements of the impacts of beneficial mutations will be instrumental in testing the validity of the Product neutrality function in this other regime. From our modeling efforts, we anticipate that such measurements could give important insights into the underlying genetic network regulating growth and proliferation.

## Methods

### Analysis of the SGA dataset

The complete SGA dataset was accessed on the cell map webpage. In the SGA genetic interaction dataset, a set of query mutant strains is crossed to an ordered array of mutants.

There are two sets of query mutants. The first one consists of a mix of nonessential deletion mutant strains and of temperature-sensitive alleles of essential genes. The second one is a set of mutants carrying hypomorphic, Decreased Abundance by mRNA Perturbation (DAmP) alleles of essential genes.

There are also two types of arrays. The Deletion Mutant Array (DMA) denotes deletions to a set of nonessential genes, while the Temperature Sensitive Array (TSA) contains a mix of essential and nonessential genes.

Both sets of query mutants are crossed to either type of array, at two different temperature conditions, namely 26 and 30°C. The analysis of *Figure 1* reports the analysis of the first set of query mutants crossed to the DMA at 30°C. In *Appendix 1—figure 1*, we report the same analysis for the other array–temperature combinations. In *Appendix 1—figure 2*, we report the analysis for the DAmP set of query mutants in the different array–temperature combinations.

For each subdataset—that is, each combination of query mutants, array, and temperature condition—the data is processed in the following steps. First, only deleterious mutations are kept. That is,

we remove mutants having a fitness larger than 1. Second, we eliminate mutants where the Additive model predicts a negative fitness, that is, such that

$$W_x + W_y < 1,$$

because in this case the prediction under the Additive neutrality function is negative ($W_x + W_y - 1 < 0$). While we could have analyzed these datapoints with the other neutrality functions, we sought to analyze all neutrality functions on the same consistent dataset.

For the sake of clarity, the scatter plots in *Figure 1* do not reproduce the entire dataset. Instead, the dataset is binned in 10 bins along the *x*-axis, and 500 values are sampled at random in that bin. However, the median lines on top of the scatter plots (e.g. *Figure 1B–D*) as well as the box plots (e.g. *Figure 1*, *Appendix 1—figures 1 and 2*) apply to the entire dataset.

## Analysis of the GO biological processes

The analysis of GO biological processes is based on the Uniprot database. In this dataset, genes are associated with a series of GO biological processes. To analyze the behavior of the fitness of double mutants associated with different biological processes, we first selected the set of biological processes that were represented by a large enough number of single mutants in the SGA dataset. Arbitrarily, this limit was set at 50. This led to a limited number of 47 biological processes (and a maximum total of 1081 pairs), which we report in the section 'Analysis of GO biological processes'. Naturally, as genes are potentially associated with multiple GO biological processes, this analysis sometimes leads to pairs of processes with genes in common. In this case, we discarded the pair so that we only consider biological process pairs that do not have any genes in common. This results in 685 pairs of biological processes that do not share any genes.

## Mutational analysis of growth models

A mutation is modeled as a perturbation of a parameter that decreases the growth rate. For a given parameter $\alpha$, we model a perturbation as $\alpha' = \theta\alpha$, where $\theta$ is a random variable uniformly distributed in $[0, 1]$. When estimating the impact of the parameter $\gamma$ in the Weiße model (see section 'Nonlinear kinetics drive deviations from the Product neutrality function in the Weiße model for cell growth'), we perform a mutational analysis as described above for different values of the parameter $\gamma$ and collect the median residual.

---

# Additional information

### Funding

| Funder | Grant reference number | Author |
| --- | --- | --- |
| National Institutes of Health | GM134858 | Jan M Skotheim |

The funders had no role in study design, data collection, and interpretation, or the decision to submit the work for publication.

### Author contributions

Lucas Fuentes Valenzuela, Conceptualization, Formal analysis, Investigation, Writing – original draft, Writing – review and editing; Paul Francois, Conceptualization, Formal analysis, Supervision, Investigation, Writing – review and editing; Jan M Skotheim, Conceptualization, Resources, Supervision, Funding acquisition, Investigation, Writing – original draft, Project administration, Writing – review and editing

### Author ORCIDs

Lucas Fuentes Valenzuela http://orcid.org/0009-0002-7403-3537
Paul Francois https://orcid.org/0000-0002-2223-839X
Jan M Skotheim https://orcid.org/0000-0001-8420-6820

Reviewer #1 (Public review): https://doi.org/10.7554/eLife.105265.3.sa1
Reviewer #2 (Public review): https://doi.org/10.7554/eLife.105265.3.sa2
Author response https://doi.org/10.7554/eLife.105265.3.sa3

## Additional files

### Supplementary files
MDAR checklist

### Data availability

The current manuscript is a computational study, so no data have been generated for this manuscript. Modeling code is uploaded to the Skotheimlab Github repository https://github.com/skotheimlab/GrowthModels (copy archived at *Fuentes Valenzuela, 2025*).

The following previously published dataset was used:

| Author(s) | Year | Dataset title | Dataset URL | Database and Identifier |
|---|---|---|---|---|
| Costanzo M, VanderSluis B, Koch E, Baryshnikova A, Pons C, Tan G, Wang W, Usaj M, Hanchard J, Lee S, Pelechano V, Styles E, Billmann M, Van Leeuwen J, Van Dyk N, Lin Z, Kuzmin E, Nelson J, Piotrowski J, Srikumar T, Bahr S, Chen Y, Deshpande R, Kurat C, Li S, Li Z, Mattiazzi Usaj M, Okada H, Pascoe N, San Luis B, Sharifpoor S, Shuteriqi E, Simpkins S, Snider J, Garadi Suresh H, Tan Y, Zhu H, Malod-Dognin N, Janjic V, Przulj N, Troyanskaya O, Stagljar I, Xia T, Ohya Y, Gingras A, Raught B, Boutros M, Steinmetz L, Moore C, Rosebrock A, Caudy A, Myers C, Andrews B, Boone C | 2025 | Data from: A global genetic interaction network maps a wiring diagram of cellular function | https://doi.org/10.5061/dryad.4291s | Dryad Digital Repository, 10.5061/dryad.4291s |

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

## Appendix 1

### A.1 Analysis of GO biological processes

The 47 biological processes that result from the analysis (with their GO code) are given in *Appendix 1—table 1*.

### A.2 Scott–Hwa model

#### A.2.1 Derivation of the double-mutant fitness

Here, we derive the expression for the double-mutant fitness in the Scott–Hwa model. First of all, the wild-type growth rate $\lambda_{WT}$ is written as

$$\lambda_{WT} = \frac{\kappa_t \kappa_n}{\kappa_t + \kappa_n}. \tag{1}$$

As we consider only deleterious mutations, we can model a perturbation to parameters as

$$\kappa_n' = (1 - \varepsilon)\kappa_n, \tag{2}$$

$$\kappa_t' = (1 - \varepsilon)\kappa_t, \tag{3}$$

where $\delta, \varepsilon \in [0, 1]$. For simplicity, we will denote by the $x$ subscript mutations affecting $\kappa_t$ and the $y$ subscript mutations affecting $\kappa_n$. We note the order does not matter. Therefore, the different fitnesses $W. = \lambda./\lambda_{WT}$ are given by

$$W_x = \frac{1 - \varepsilon}{1 - \varepsilon \kappa_t/(\kappa_t + \kappa_n)}, \tag{4}$$

$$W_y = \frac{1 - \delta}{1 - \delta \kappa_n/(\kappa_t + \kappa_n)}, \tag{5}$$

$$W_{xy} = \frac{(1 - \varepsilon)(1 - \delta)(\kappa_t + \kappa_n)}{(1 - \varepsilon)\kappa_t + (1 - \delta)\kappa_n}. \tag{6}$$

From there, we can calculate that

$$W_x W_y = \frac{(1 - \varepsilon)(1 - \delta)(\kappa_t + \kappa_n)}{(1 - \varepsilon)\kappa_t + (1 - \delta)\kappa_n + \varepsilon\delta\lambda_{WT}}. \tag{7}$$

This expression is similar to *Equation 6*. Indeed, we can rewrite it as

$$W_{xy} = W_x W_y \left( 1 + \frac{\varepsilon\delta\lambda_{WT}}{(1 - \varepsilon)\kappa_t + (1 - \delta)\kappa_n} \right). \tag{8}$$

We see that the double-mutant fitness $W_{xy}$ consists of the product of single-mutant fitnesses and a deviation. We will now express this deviation as a function of single-mutant fitnesses only, showing that the expression does not depend on the value of the parameters $\kappa_t$ and $\kappa_n$.

Rearranging *Equations 4 and 5*, we have

$$1 - W_x = \frac{\varepsilon\kappa_n}{(1 - \varepsilon)\kappa_t + \kappa_n}, \tag{9}$$

$$1 - W_y = \frac{\delta\kappa_t}{(1 - \delta)\kappa_n + \kappa_t}, \tag{10}$$

$$\Rightarrow (1 - W_x)(1 - W_y) = \frac{1}{\kappa_t + \kappa_n} \frac{\varepsilon\delta\lambda_{WT}}{(1 - \varepsilon\kappa_t/(\kappa_t + \kappa_n))(1 - \delta\kappa_n/(\kappa_t + \kappa_n))}. \tag{11}$$

which can be rearranged to see that

$$\frac{\varepsilon\delta\lambda_{WT}}{(1 - \varepsilon)\kappa_t + (1 - \delta)\kappa_n} = (1 - W_x)(1 - W_y)\frac{(1 - \varepsilon\kappa_t/(\kappa_t + \kappa_n))(1 - \delta\kappa_n/(\kappa_t + \kappa_n))}{(1 - \varepsilon)\kappa_t + (1 - \delta)\kappa_n}, \tag{12}$$

$$= (1 - W_x)(1 - W_y) \frac{1 - \varepsilon\kappa_t/(\kappa_t + \kappa_n) - \delta\kappa_n/(\kappa_t + \kappa_n) + \varepsilon\delta\lambda_{WT}/(\kappa_t + \kappa_n)}{(1 - \varepsilon)\kappa_t + (1 - \delta)\kappa_n}, \tag{13}$$

$$= (1 - W_x)(1 - W_y) \frac{(1 - \varepsilon)\kappa_t + (1 - \delta)\kappa_n + \varepsilon\delta\lambda_{WT}}{(1 - \varepsilon)\kappa_t + (1 - \delta)\kappa_n}, \tag{14}$$

$$= (1 - W_x)(1 - W_y) \left( 1 + \frac{\varepsilon\delta\lambda_{WT}}{(1 - \varepsilon)\kappa_t + (1 - \delta)\kappa_n} \right). \tag{15}$$

We see a recursive relationship appear, such that

$$\frac{\varepsilon\delta\lambda_{WT}}{(1 - \varepsilon)\kappa_t + (1 - \delta)\kappa_n} = (1 - W_x)(1 - W_y) \left( 1 + (1 - W_x)(1 - W_y) \left( 1 + \dots \right) \right), \tag{16}$$

$$= (1 - W_x)(1 - W_y) + \left( (1 - W_x)(1 - W_y) \right)^2 + \left( (1 - W_x)(1 - W_y) \right)^3 + \dots \tag{17}$$

Therefore, we have from **Equation 8** that

$$W_{xy} = W_x W_y \sum_{k=0}^{\infty} \left( (1 - W_x)(1 - W_y) \right)^k, \tag{18}$$

$$= W_x W_y \frac{1}{1 - (1 - W_x)(1 - W_y)}, \tag{19}$$

$$= W_x W_y \left( 1 + \frac{(1 - W_x)(1 - W_y)}{1 - (1 - W_x)(1 - W_y)} \right), \tag{20}$$

from the property of geometric series and noting that $(1 - W_x)(1 - W_y) \leq 1$.

### A.2.2 Scott–Hwa model with no feedback

To understand the impact of growth rate optimization in the Scott–Hwa model, we consider an alternative model formulation where this feedback is absent. We assume that we still have two different processes (metabolism and translation) and that the flux through both of them has to be equal. However, we do not incorporate the assumption that the proteome has to be partitioned between these two processes. Instead, each process is assumed to have a given amount of proteins or enzymes available at its disposal. This formulation implies that

$$\lambda = \kappa_t \phi_t = \kappa_n \phi_n, \tag{21}$$

$$0 \leq \phi_t \leq \phi_t^{max}, \tag{22}$$

$$0 \leq \phi_n \leq \phi_n^{max}, \tag{23}$$

where $\kappa_t$ and $\kappa_n$ are the capacities of the translation and metabolic sector, respectively, as in the Scott–Hwa model, and where $\phi_t$ and $\phi_n$ are the normalized concentrations of proteins or enzymes.

If we assume that the cell maximizes its growth rate, we have that

$$\lambda_{max}^{nf} = \min(\kappa_t \phi_t^{max}, \kappa_n \phi_n^{max}), \tag{24}$$

where the $nf$ superscript stands for *no feedback*.

Clearly, in this simplified model, one sector will be limiting either because it is not efficient enough, that is, $\kappa_i$ is too small, or because it does not have enough resources to deploy to accelerate the process, that is, $\phi_i^{max}$ is too small. In this case, we see that the growth rate $\lambda$ corresponds to the minimum of the potential maximal fluxes in either sector.

Therefore, modeling mutations as affecting either $\kappa_t$ or $\kappa_n$ but not the protein fractions, we find that

$$W_x = \frac{\min(\kappa_t'\phi_t^{max}, \kappa_n\phi_n^{max})}{\lambda_0},$$

$$W_y = \frac{\min(\kappa_t\phi_t^{max}, \kappa_n'\phi_n^{max})}{\lambda_0},$$

$$W_{xy} = \frac{\min(\kappa_t'\phi_t^{max}, \kappa_n'\phi_n^{max})}{\lambda_0}.$$

For the sake of argument, assume that $\kappa_t\phi_t^{max} \leq \kappa_n\phi_n^{max}$. Therefore, $\lambda_0 = \kappa_t\phi_t^{max}$ and $W_x = \frac{\kappa_t'\phi_t^{max}}{\lambda_0} = \frac{\kappa_t'}{\kappa_t}$, and $W_y = \min(1, \frac{\kappa_n'\phi_n^{max}}{\kappa_t\phi_t^{max}})$.

We can then write that

$$W_{xy} = \min\left(W_x, \frac{\kappa_n'\phi_n^{max}}{\kappa_t\phi_t^{max}}\right).$$

If $\frac{\kappa_n'\phi_n^{max}}{\kappa_t\phi_t^{max}} > 1$, then $W_{xy} = W_x \leq 1$. However, if $\frac{\kappa_n'\phi_n^{max}}{\kappa_t\phi_t^{max}} \leq 1$, then $W_y = \frac{\kappa_n'\phi_n^{max}}{\kappa_t\phi_t^{max}}$. Therefore, we have that

$$W_{xy} = \min\left(W_x, W_y\right),$$

that is, in the model with no feedback laid out above, double-mutant fitnesses are actually the minimum of the single-mutant fitnesses. Simulation results are reported in *Appendix 1—figure 4*.

We note that this absence of feedback results in significantly smaller growth rates. In the Scott–Hwa model, we have

$$\lambda^{SH} = \frac{\kappa_t\kappa_n}{\kappa_t + \kappa_n}.$$

This implies that $\phi_t = \frac{\kappa_n}{\kappa_t+\kappa_n}$ and $\phi_n = \frac{\kappa_t}{\kappa_t+\kappa_n}$.

Let us now consider, in the context of the model in this section, $\phi_t^{max} = \frac{\kappa_n}{\kappa_t+\kappa_n}$ and $\phi_n^{max} = \frac{\kappa_t}{\kappa_t+\kappa_n}$. Upon mutation of $\kappa_t$ or $\kappa_n$, we have

$$\frac{\lambda^{nf}}{\lambda^{SH}} = \frac{\min\left(\kappa_t'\dfrac{\kappa_n}{\kappa_t+\kappa_n}, \kappa_n'\dfrac{\kappa_t}{\kappa_t+\kappa_n}\right)}{\dfrac{\kappa_t'\kappa_n'}{\kappa_t'+\kappa_n'}} \tag{25}$$

$$= \min\left(\frac{\kappa_n(\kappa_t'+\kappa_n')}{\kappa_n'(\kappa_t+\kappa_n)}, \frac{\kappa_t(\kappa_t'+\kappa_n')}{\kappa_t'(\kappa_t+\kappa_n)}\right), \tag{26}$$

$$= \frac{\kappa_t'+\kappa_n'}{\kappa_t+\kappa_n}\min\left(\frac{\kappa_n}{\kappa_n'}, \frac{\kappa_t}{\kappa_t'}\right). \tag{27}$$

Assume, without loss of generality, that $\frac{\kappa_t}{\kappa_t'} \leq \frac{\kappa_n}{\kappa_n'}$, that is $\frac{\kappa_n'}{\kappa_n} \leq \frac{\kappa_t'}{\kappa_t}$. Then, we have

$$\frac{\kappa_t'+\kappa_n'}{\kappa_t+\kappa_n} \leq \frac{\kappa_t'+\kappa_t'\dfrac{\kappa_n}{\kappa_t}}{\kappa_t+\kappa_n} = \frac{\kappa_t'}{\kappa_t},$$

which implies that $\lambda^{nf}/\lambda^{SH} \leq 1$. Therefore, in the absence of feedback, the growth rate is smaller than in the Scott–Hwa model, which is consistent with the interpretation of growth-rate optimization.

## A.3 Weiße model
### A.3.1 Original model
The original model consists of a system of equations describing the synthesis, degradation, and reaction of a series of molecules and proteins. In particular, external nutrients $s$ are converted into internal nutrients $s_i$. Those nutrients are then converted into a generic cellular energy $a$ that enables transcription and translation. Indeed, mRNAs $m$ are transcribed and then bind to ribosomes to form an mRNA–ribosome complex $c$ that governs the rate of protein translation. There are four main types of mRNAs and complexes in the model, indicated by a subscript. Each is associated

with different processes: transport, metabolism, ribosomal, and q-proteins. This last type of protein denotes housekeeping proteins whose concentration remains approximately constant.

$$\dot{s}_i = \nu_{\text{imp}}(e_t, s) - \nu_{\text{cat}}(e_m, s_i) - \lambda s_i, \tag{28}$$

$$\dot{a} = n_s \cdot \nu_{\text{cat}}(e_m, s_i) - \sum_{x \in \{r,t,m,q\}} n_x \nu_x(c_x, a) - \lambda a, \tag{29}$$

$$\dot{r} = \nu_r(c_r, a) - \lambda r + \sum_{x \in \{r,t,m,q\}} \left( \nu_x(c_x, a) - k_b r m_x + k_u c_x \right), \tag{30}$$

$$\dot{e}_t = \nu_t(c_t, a) - \lambda e_t, \tag{31}$$

$$\dot{e}_m = \nu_m(c_m, a) - \lambda e_m, \tag{32}$$

$$\dot{q} = \nu_q(c_q, a) - \lambda q, \tag{33}$$

$$\dot{m}_x = \omega_x(a) - (\lambda + d_m)m_x + \nu_x(c_x, a) - k_b r m_x + k_u c_x, \tag{34}$$

$$\dot{c}_x = -\lambda c_x + k_b r m_x - k_u c_x - \nu_x(c_x, a). \tag{35}$$

All of these reactions are modulated by parameters reported in *Appendix 1—table 2*.

The rates governing the system of equations are assumed to follow Michaelis–Menten kinetics as follows

$$\nu_{\text{imp}}(e_t, s) = e_t \frac{v_t s}{K_t + s}, \tag{36}$$

$$\nu_{\text{cat}}(e_m, s_i) = e_m \frac{v_m s_i}{K_m + s_i}, \tag{37}$$

$$\nu_x(c_x, a) \sim c_x \frac{\gamma(a)}{n_x}, \tag{38}$$

$$\gamma(a) := \frac{\gamma_{max} a}{K_\gamma + a}, \tag{39}$$

$$\omega_x(a) = \omega_x \frac{a}{\theta_x + a}. \tag{40}$$

Finally, the growth rate corresponds to total mass of proteins being synthesized at steady state, that is

$$\lambda = \frac{\gamma(a) \sum_x c_x}{M}. \tag{41}$$

## A.3.2 Isolation of parameters

This model has a total of 21 parameters whose values were set (either from the literature or estimated) in the original model (Table S2 in *Weiße et al., 2015*; reproduced here under *Appendix 1—table 2*).

Among them, some represent quantities or parameters that would not reasonably change under a mutation. Therefore, we do not consider the following parameters in our mutation analysis: external nutrients $s$, mRNA degradation rate $d_m$, ribosome length $n_r$, length of non-ribosomal proteins $n_x$, q-autoinhibition Hill coefficient $h_q$, the mRNA–ribosome binding/unbinding rates $k_b$ and $k_u$, the total cell mass $M$.

To determine whether a numerical mutational analysis with the remaining 13 parameters could be considered, we computed the impact of a change of these parameters on the growth rate $\lambda$ (*Appendix 1—figure 6*). Upon mutation, most parameters do impact the growth rate negatively. However, two parameters $v_m, K_m$ do not seem to have an impact on the growth rate. We attribute this to the metabolic sector not being limiting in this model, with those parameter values. We, therefore,

exclude those two parameters from further analysis. We also exclude the parameters associated with the q-proteins $K_q$ and $\omega_q$ as these are associated mostly with housekeeping functions.

We are therefore left with nine parameters modeling the impact of four different processes. In total, this results in 28 potential combinations of two parameters associated with different processes.

### A.3.3 Simplification

To facilitate analytical treatment, we consider a simplified model with only two types of proteins, instead of four: a general protein $p$ and ribosomes $r$. We also eliminate the equations corresponding to the competitive binding of mRNAs for ribosomes. Practically, this assumption means that every mRNA binds immediately to a ribosome. While this assumption might not be true in all regimes, we find that this simplifies the analytical treatment and still enables us to understand something concrete about the model.

The simplified model we consider is

$$\dot{s}_i = \nu_{\text{imp}}(p, s) - \nu_{\text{cat}}(p, s_i) - \lambda s_i, \tag{42}$$

$$\dot{a} = n_s \nu_{\text{cat}}(p, s_i) - \sum_x n_x \nu_x(c_p, a) - \lambda a, \tag{43}$$

$$\dot{r} = \nu_r(c_r, a) - \lambda r + \sum_x \nu_x(c_x, a), \tag{44}$$

$$\dot{p} = \nu_p(p, a) - \lambda p, \tag{45}$$

$$\dot{c}_r = \omega_r(a) - \nu_r(c_r, a) - \lambda c_r, \tag{46}$$

$$\dot{c}_p = \omega_p(a) - \nu_p(c_p, a) - \lambda c_p. \tag{47}$$

If we assume the dilution fluxes to be negligible in comparison to other fluxes in **Equations 42–47**, we get that

$$\nu_{\text{imp}}(p, s) \sim \nu_{\text{cat}}(p, s_i) \Rightarrow p\nu_t \frac{s}{K_t + s} \sim p\nu_m \frac{s_i}{K_m + s_i}, \tag{48}$$

$$n_s \nu_{\text{cat}}(p, s_i) \sim \sum_x n_x \nu_x(c_p, a) = (c_p + c_r)\gamma(a), \tag{49}$$

$$\omega_p(a) \sim \nu_p(c_p, a), \tag{50}$$

$$\omega_r(a) \sim \nu_r(c_r, a). \tag{51}$$

The dilution flux needs to be nonzero for $p$, as its governing equation only contains two terms. We have therefore that

$$\nu_p(p, a) = \lambda p \Rightarrow p \sim \omega_p \frac{a}{\theta_x + a} \frac{1}{\lambda}. \tag{52}$$

As the growth rate $\lambda = \frac{\gamma(a) \sum_x c_x}{M}$, we have

$$\lambda = \frac{n_s \nu_{\text{cat}}}{M} \sim \frac{1}{M} n_s \nu_t p \frac{s}{K_t + s}. \tag{53}$$

Injecting **Equation 52** into **Equation 53**, we have

$$\lambda \sim \sqrt{\frac{1}{M} n_s \nu_t w_p \frac{s}{K_t + s} \frac{a}{\theta_x + a}}. \tag{54}$$

From the above expression, we see that terms belonging to different sectors combine as a product. However, the energy $a$ is **not** a parameter, but rather a variable in the model. Its value also directly depends on the values of other parameters.

From **Equation 49**, we have that

$$\lambda \sim \frac{1}{M} \left( n_p \omega_p \frac{a}{\theta_x + a} + n_r \omega_r \frac{a}{\theta_r + a} \right). \tag{55}$$

If $\theta_x < a < \theta_r$, we can approximate that with

$$\lambda \sim \frac{1}{M} \left( n_p \omega_p + n_r \omega_r \frac{a}{\theta_r} \right). \tag{56}$$

If $a \ll \theta_x, \theta_r$, we have

$$\lambda \sim \frac{1}{M} \left( \frac{n_p \omega_p}{\theta_x} + \frac{n_r \omega_r}{\theta_r} \right) a. \tag{57}$$

In both cases, we can assume that there is a linear relationship between the growth rate $\lambda$ and the energy vector $a$, such that

$$a \sim b\lambda + c. \tag{58}$$

We can inject that in the $a/(\theta_x + a)$ term in **Equation 54**, which gives

$$\frac{a}{\theta_x + a} \sim \frac{\lambda + c/b}{(\theta_x + c)/b + \lambda}. \tag{59}$$

Assuming that $c/b \ll \lambda$, we have

$$\frac{a}{\theta_x + a} \sim \frac{\lambda}{K' + \lambda}, \tag{60}$$

with $K' = (\theta_x + c)/b$.

Injecting the above in **Equation 54**, we get that,

$$\lambda \sim \sqrt{\frac{1}{M} n_s v_t w_p \frac{s}{K_t + s} \frac{\lambda}{K' + \lambda}}. \tag{61}$$

Therefore, $K'$ acts as a scale of growth rate where deviations from the Product model can appear. If $K' \gg \lambda$, then

$$\lambda \sim \frac{1}{M} n_s v_t w_p \frac{s}{K_t + s}. \tag{62}$$

On the other hand, if $K' \ll \lambda$, then

$$\lambda \sim \sqrt{\frac{1}{M} n_s v_t w_p \frac{s}{K_t + s}}. \tag{63}$$

In both of these cases, the growth rate $\lambda$ behaves as the product of multiple parameters. However, if $\lambda \sim K'$, and if we denote two parameters in the expression by $\alpha$, $\beta$ without loss of generality, we have

$$\lambda^2(K' + \lambda) \sim \alpha\beta\lambda. \tag{64}$$

Either $\lambda = 0$ (which we will not consider), or

$$
\begin{aligned}
\lambda(K' + \lambda) &= \alpha\beta \\
\Leftrightarrow \lambda^2 + K'\lambda - \alpha\beta &= 0 \\
\Leftrightarrow \lambda &= \frac{-K' + \sqrt{K'^2 + 4\alpha\beta}}{2} \\
&= \frac{K'}{2} \left( -1 + \sqrt{1 + \frac{4\alpha\beta}{K'^2}} \right).
\end{aligned}
$$

We can write, for small $x$, that

$$\sqrt{1+x} = 1 + \frac{x}{2} - \frac{x^2}{8} + \dots$$

Assuming that $\frac{4\alpha\beta}{K'^2}$ is small and keeping terms up to second order, we therefore have that

$$\lambda = \frac{K'}{2}\frac{x}{2}\left(1 - \frac{x}{4}\right), \text{ with } x = \frac{4\alpha\beta}{K'^2}$$
$$\Rightarrow \lambda \propto \alpha\beta\left(1 - \gamma\alpha\beta\right), \text{ with } \gamma = 1/K'^2.$$

## A.4 Double-mutant fitness

From there, we can compute an approximation to the double-mutant fitness. Let us assume that $\lambda = \alpha\beta\left(1 - \gamma\alpha\beta\right)$ as derived above. To simplify notation, we will rescale the parameters to their *WT* value, that is

$$\alpha' = \alpha/\alpha_{WT},$$
$$\beta' = \beta/\beta_{WT},$$
$$\bar{\gamma} = \gamma\alpha_{WT}\beta_{WT}.$$

Under this rescaling, we see that

$$\lambda_{WT} = \alpha_{WT}\beta_{WT}(1 - \bar{\gamma}),$$
$$W_x = \frac{\alpha'(1 - \bar{\gamma}\alpha')}{1 - \bar{\gamma}},$$
$$W_y = \frac{\beta'(1 - \bar{\gamma}\beta')}{1 - \bar{\gamma}},$$
$$W_{xy} = \alpha'\beta'\frac{1 - \bar{\gamma}\alpha'\beta'}{(1 - \bar{\gamma})}.$$

Noting that

$$W_x W_y = \frac{\alpha'\beta'(1 - \bar{\gamma}\alpha')(1 - \bar{\gamma}\beta')}{(1 - \bar{\gamma})^2},$$

we can show that

$$W_{xy} = W_x W_y - \frac{\alpha'\beta'}{(1 - \bar{\gamma})^2}\bar{\gamma}(1 - \alpha')(1 - \beta'),$$
$$= W_x W_y \left(1 - \bar{\gamma}\frac{1 - \alpha'}{1 - \bar{\gamma}\alpha'}\frac{1 - \beta'}{1 - \bar{\gamma}\beta'}\right).$$

If $\bar{\gamma}$ is small, then $W_x = \frac{\alpha'(1 - \bar{\gamma}\alpha')}{1 - \bar{\gamma}} \sim \alpha'$ and $W_y = \frac{\beta'(1 - \bar{\gamma}\beta')}{1 - \bar{\gamma}} \sim \beta'$, and therefore we can write that

$$W_{xy} \sim W_x W_y \left(1 - \bar{\gamma}\frac{1 - W_x}{1 - \bar{\gamma}W_x}\frac{1 - W_y}{1 - \bar{\gamma}W_y}\right).$$

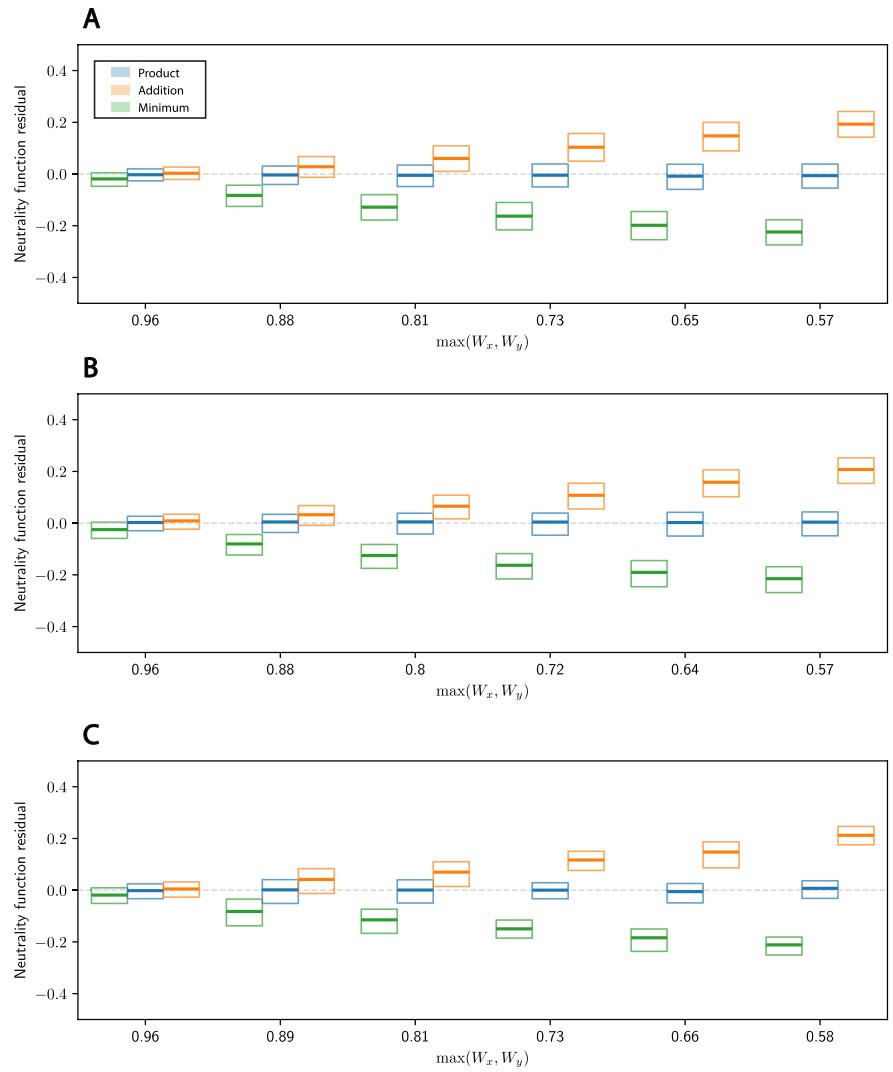

**Appendix 1—figure 1.** Double-mutant fitnesses are best described by the Product neutrality function in the Synthetic Genetic Array (SGA) dataset. Box plots for the distributions of the residuals for the three neutrality functions as a function of the maximum single-mutant fitness. Each plot corresponds to a different subset of the SGA dataset. Namely, they correspond to the first set of query mutants (see Methods) crossed to different types of mutant arrays in different temperature conditions (**A**) Deletion Mutant Array at 26°C. (**B**) Temperature Sensitive Array at 26°C. (**C**) Temperature Sensitive Array at 30°C. Thick line denotes the median, and boxes denote the 25th and 75th percentiles of the distributions. The Product neutrality function models the data consistently better than the others.

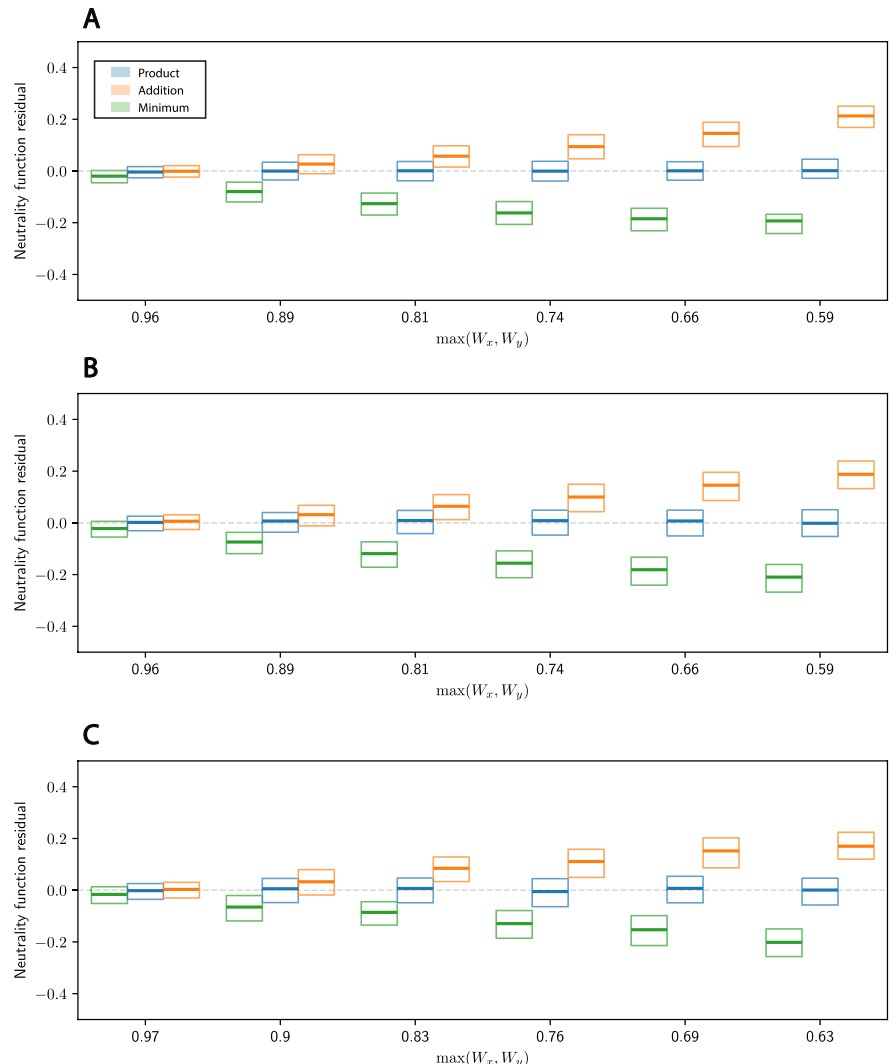

**Appendix 1—figure 2.** Double-mutant fitnesses are best described by the Product neutrality function in the Synthetic Genetic Array (SGA) dataset. Box plots for the distributions of the residuals for the three neutrality functions as a function of the maximum single-mutant fitness. Each plot corresponds to a different subset of the SGA dataset. Namely, they correspond to the second set of query mutants (DAmP, see Methods) crossed to different types of mutant arrays in different temperature conditions (**A**) Deletion Mutant Array at 30°C. (**B**) Temperature Sensitive Array at 26°C. (**C**) Temperature Sensitive Array at 30°C. Thick line denotes the median, and boxes denote the 25th and 75th percentiles of the distributions. The Product neutrality function models the data consistently better than the others.

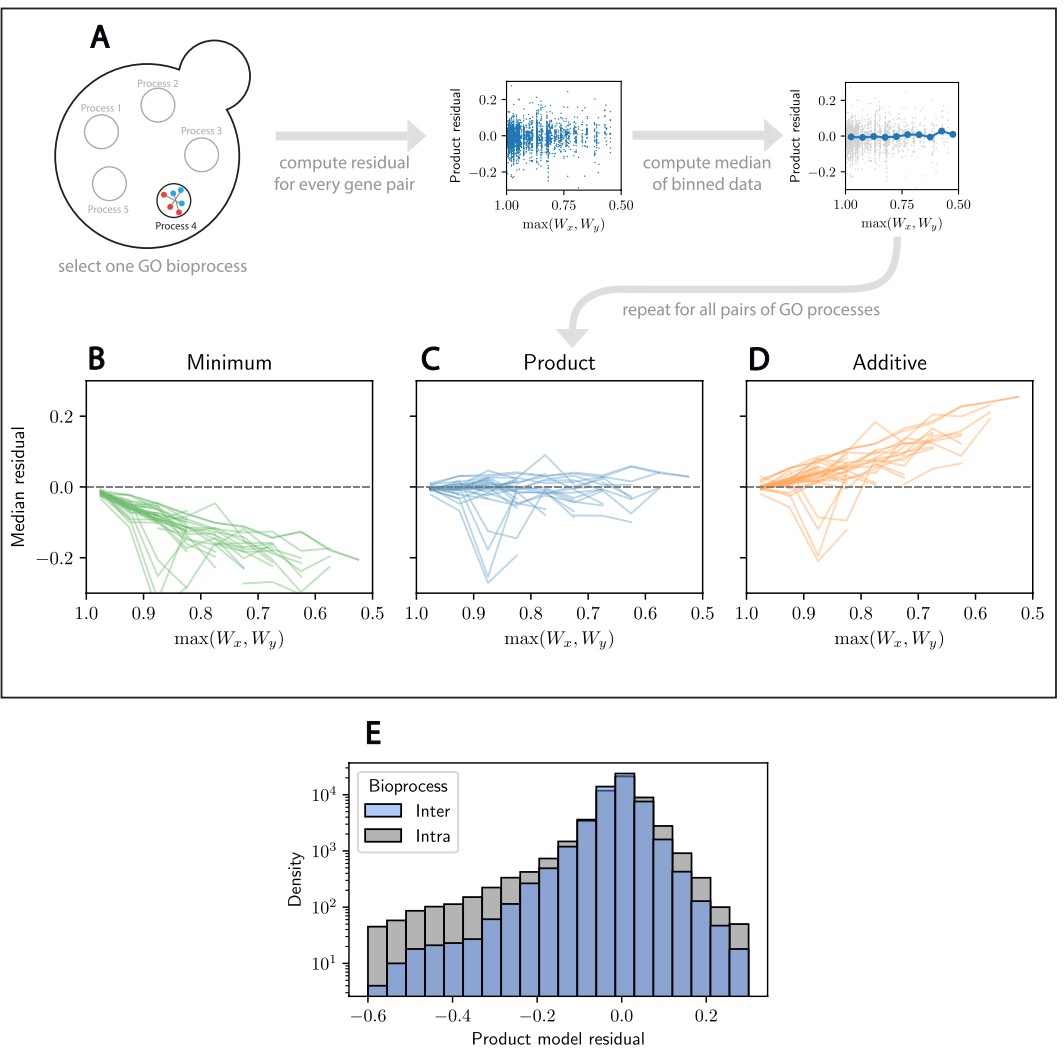

**Appendix 1—figure 3.** Larger deviations from the Product neutrality function characterize gene pairs affecting the same GO biological process. (**A**) Schematic illustration of the analysis process for double mutants where both mutations affect the same GO biological process. We first select two different GO biological processes and extract the double mutants in the Synthetic Genetic Array (SGA) dataset associated with them. Then, we compute the median residual for each pair of biological processes and each neutrality function. (**B–D**) Median residual for the Minimum, Product, and Additive neutrality functions as a function of the maximum single-mutant fitness. Each line denotes mutations to a single GO biological process. We see larger deviations from the Product model than in *Figure 2*. (**E**) Histogram of the SGA dataset after extracting pairs affecting either two different (inter) or the same (intra) GO biological process. Large residuals are much more likely when both mutations affect the same GO biological process.

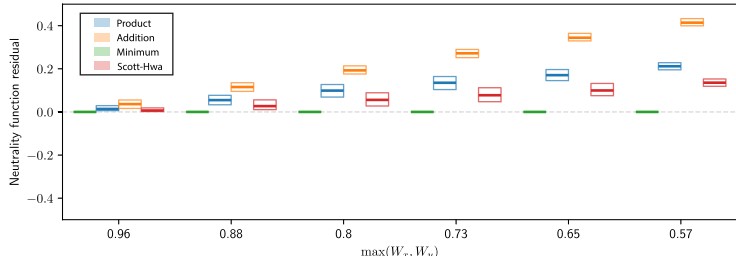

**Appendix 1—figure 4.** The Scott–Hwa model with no feedback follows a Minimum neutrality function. Box plots for the distributions of the residuals for the model of Scott–Hwa model with no feedback as a function of the maximum single-mutant fitness. A thick line denotes the median, and boxes denote the upper and lower quartiles of the data. The absence of feedback due to resource competition in the model of Scott–Hwa model with no feedback results in a Minimum neutrality function.

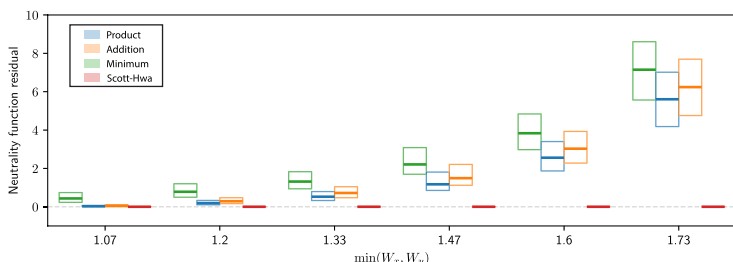

**Appendix 1—figure 5.** Large deviations from the Product neutrality function characterize beneficial mutations. Box plots for the distributions of the residuals for the different neutrality functions for beneficial mutations as a function of the minimum single-mutant fitness. Thick lines denote the median, and boxes denote the upper and lower quartiles of the data. Deviations from the Product neutrality function derived in Derivation of the double-mutant fitness can be unbounded.

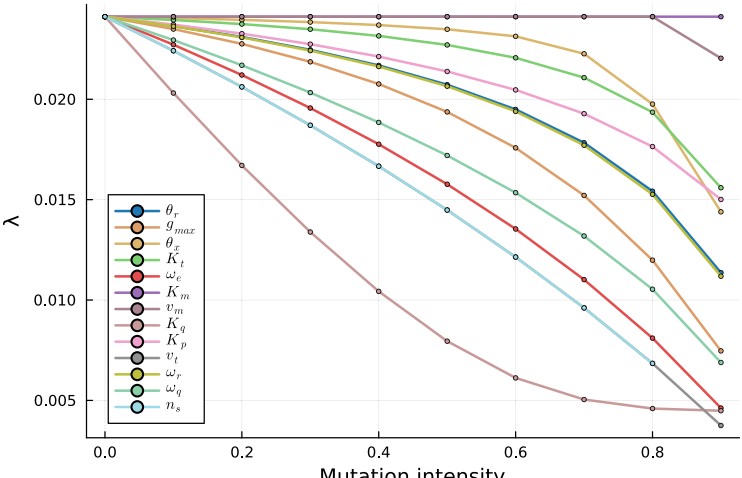

**Appendix 1—figure 6.** Eleven parameters exhibit negative impact on growth rate upon mutation in the Weiße model. Among the initial 21 parameters, 13 were kept as candidates for a mutational analysis. Two of them $v_m, K_m$, associated with the metabolic sector, do not have any impact on growth rate upon mutation, likely because this sector is not limiting for growth in that parameter range. The others have a negative impact upon mutation.

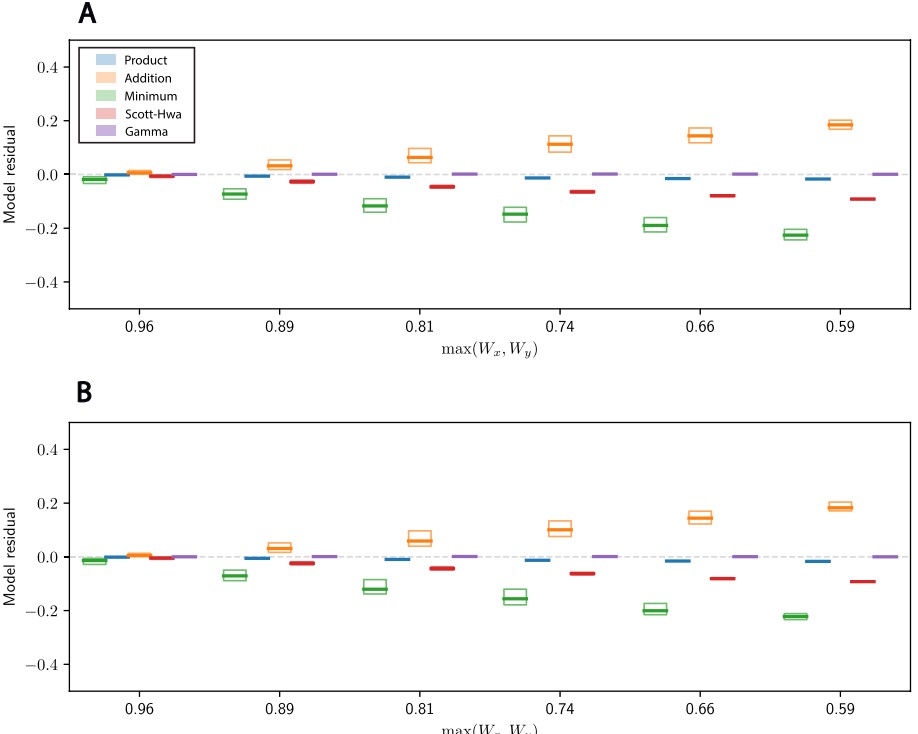

**Appendix 1—figure 7.** Deviations from the Product neutrality function in the Weiße model are captured by the γ approximation. Box plots for the distributions of the residuals for all models considered in this paper, for two example parameter pairs (**A:** $\omega_q, n_s$, **B:** $K_t, \omega_e$). A thick line denotes the median, and boxes denote the upper and lower quartiles of the data. The Gamma model, in purple, denotes the derivation in **Figure 5A**. It captures the small deviations from the Product neutrality function.

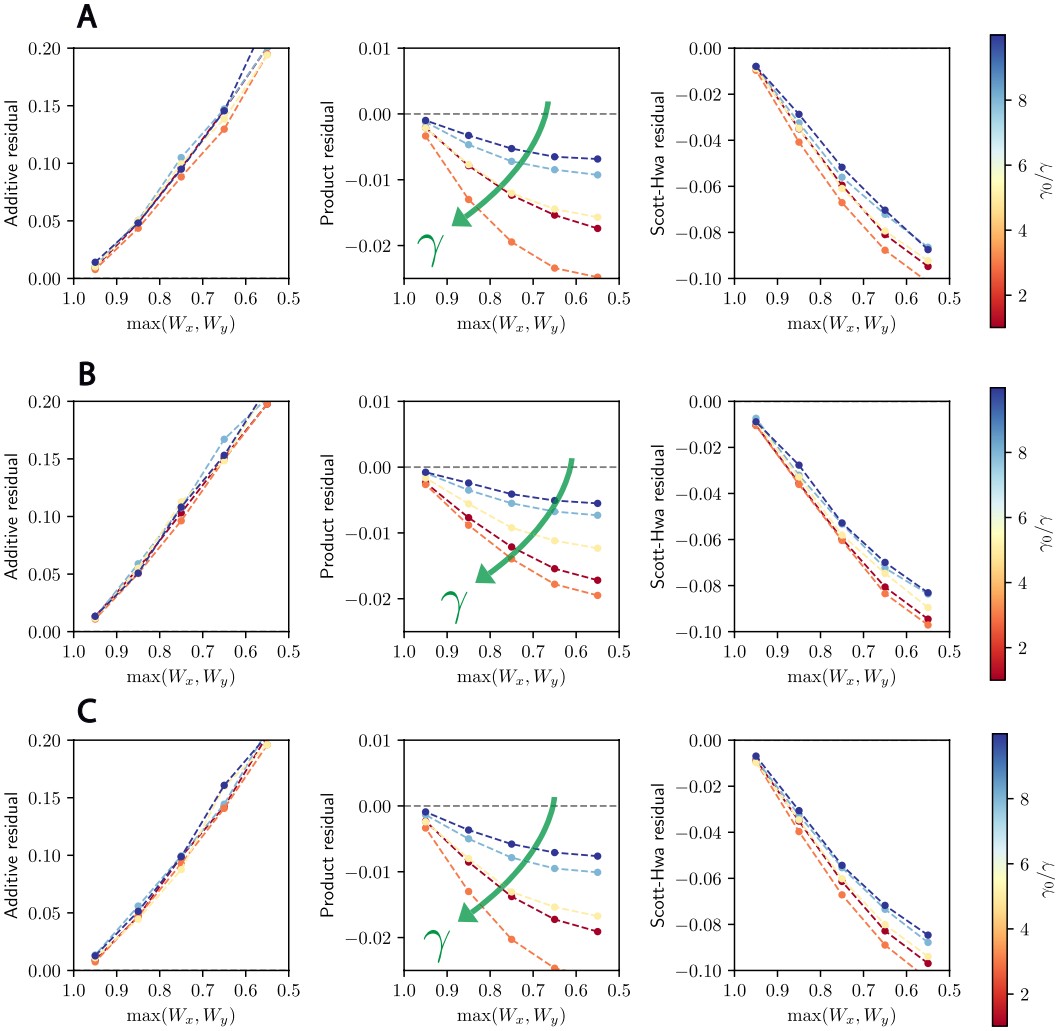

**Appendix 1—figure 8.** Tuning γ impacts how good an approximation the Product neutrality function is for multiple parameter pairs in the Weiße model. The analysis of *Figure 5C*, illustrating the impact of γ on a single parameter pair $(n_s, v_t)$ is here extended to other parameter pairs to demonstrate the validity of the mechanistic interpretation. (**A**) $v_t, \omega_r$; (**B**) $\omega_e, n_s$; (**C**) $\theta_r, v_t$ In all cases, we see that decreasing γ results in better alignment with the Product model.

**Appendix 1—table 1.** GO biological processes used in the analysis.

| GO biological process name | GO biological identifier |
| --- | --- |
| DNA integration | GO:0015074 |
| DNA recombination | GO:0006310 |
| DNA repair | GO:0006281 |
| Ascospore formation | GO:0030437 |
| Cell cycle | GO:0007049 |
| Cell division | GO:0051301 |
| Cell wall organization | GO:0071555 |
| Cellular response to DNA damage stimulus | GO:0006974 |
| Cellular response to oxidative stress | GO:0034599 |

*Appendix 1—table 1 Continued on next page*

*Appendix 1—table 1 Continued*

| GO biological process name | GO biological identifier |
|---|---|
| Chromatin remodeling | GO:0006338 |
| Chromatin silencing at telomere | GO:0006348 |
| Chromosome segregation | GO:0007059 |
| Cytoplasmic translation | GO:0002181 |
| Endocytosis | GO:0006897 |
| Endoplasmic reticulum to Golgi vesicle-mediated transport | GO:0006888 |
| Fungal-type cell wall organization | GO:0031505 |
| Intracellular protein transport | GO:0006886 |
| Intracellular signal transduction | GO:0035556 |
| mRNA splicing, via spliceosome | GO:0000398 |
| Macroautophagy | GO:0016236 |
| Maturation of SSU-rRNA from tricistronic rRNA transcript | GO:0000462 |
| Meiotic cell cycle | GO:0051321 |
| Mitochondrial translation | GO:0032543 |
| Negative regulation of transcription by RNA polymerase II | GO:0000122 |
| Positive regulation of transcription by RNA polymerase II | GO:0045944 |
| Proteasome-mediated ubiquitin-dependent protein catabolic process | GO:0043161 |
| Protein folding | GO:0006457 |
| Protein import into nucleus | GO:0006606 |
| Protein phosphorylation | GO:0006468 |
| Protein targeting to vacuole | GO:0006623 |
| Protein transport | GO:0015031 |
| Protein ubiquitination | GO:0016567 |
| Pseudohyphal growth | GO:0007124 |
| rRNA methylation | GO:0031167 |
| rRNA processing | GO:0006364 |
| Reciprocal meiotic recombination | GO:0007131 |
| Regulation of transcription by RNA polymerase II | GO:0006357 |
| Regulation of transcription, DNA-templated | GO:0006355 |
| Ribosomal large subunit biogenesis | GO:0042273 |
| Sporulation resulting in formation of a cellular spore | GO:0030435 |
| Transcription by RNA polymerase II | GO:0006366 |
| Transcription elongation from RNA polymerase II promoter | GO:0006368 |
| Translational termination | GO:0006415 |
| Transmembrane transport | GO:0055085 |
| Transposition, RNA-mediated | GO:0032197 |
| Ubiquitin-dependent protein catabolic process | GO:0006511 |
| Vesicle-mediated transport | GO:0016192 |

**Appendix 1—table 2.** Model parameters from *Weiße et al., 2015*, obtained either from the literature or from parameter optimization.

| | Description | Default value | Unit |
|---|---|---|---|
| $s$ | External nutrient | $10^4$ | [molecs] |
| $d_m$ | mRNA-degradation rate | 0.1 | [min$^{-1}$] |
| $n_s$ | Nutrient efficiency | 0.5 | None |
| $n_r$ | Ribosome length | 7459 | [aa/molecs] |
| $n_x, x \in \{t, m, q\}$ | Length of non-ribosomal proteins | 300 | [aa/molecs] |
| $\gamma_{max}$ | Max. transl. elongation rate | 1260 | [aa/min molecs] |
| $K_\gamma$ | Transl. elongation threshold | 7 | [molecs/cell] |
| $v_t$ | Max. nutrient import rate | 726 | [min$^{-1}$] |
| $K_t$ | Nutrient import threshold | 1000 | [molecs] |
| $v_m$ | Max. enzymatic rate | 5800 | [min$^{-1}$] |
| $K_m$ | Enzymatic threshold | 1000 | [molecs/cell] |
| $w_r$ | Max. ribosome transcription rate | 930 | [molecs/min cell] |
| $w_e = w_t = w_m$ | Max. enzyme transcription rate | 4.14 | [molecs/min cell] |
| $w_q$ | Max. $q$-transcription rate | 948.93 | [molecs/min cell] |
| $\theta_r$ | Ribosome transcription threshold | 426.87 | [molecs/cell] |
| $\theta_{nr}$ | Non-ribosomal transcription threshold | 4.38 | [molecs/cell] |
| $K_q$ | $q$-Autoinhibition threshold | 152,219 | [molecs/cell] |
| $h_q$ | $q$-Autoinhibition Hill coeff. | 4 | None |
| $k_b$ | mRNA–ribosome binding rate | 1 | [cell/min molecs] |
| $k_u$ | mRNA–ribosome unbinding rate | 1 | [min$^{-1}$] |
| $M$ | Total cell mass | $10^8$ | [aa] |

