## [Editor Report · eLife Assessment]

The paper addresses the question of gene epistasis and asks what is the correct null model for which we should declare no epistasis. By reanalyzing synthetic gene array datasets regarding single and double-knockout yeast mutants, and considering two theoretical models of cell growth, the authors reach the **valuable** conclusion that the product function is a good null model. While the justification of some assumptions is **incomplete**, the results have the potential to be of value to the field of gene epistasis.

---

## [Referee Report · Reviewer #1 (Public review)]

Summary

Detecting unexpected epistatic interactions between multiple mutations requires a robust null expectation-or neutral function-that predicts the combined effects of multiple mutations on phenotype based on the individual effects of single mutations. This study evaluated the relevance of the product neutrality function, where double-mutant fitness is represented as a multiplicative combination of single-mutant fitness in the absence of epistatic interactions. The authors used a recent large dataset on fitness, specifically yeast colony size, to analyze epistatic interactions.

The study confirmed that the product function outperformed other neutral functions in predicting double-mutant fitness, showing no bias between negative and positive epistatic interactions. Additionally, in the theoretical portion of the study, the authors employed a previously established theoretical model of bacterial cell growth to simulate growth rates of both single- and double-mutants under multiple parameters. The simulations similarly demonstrated that the product function was superior to other functions in predicting the fitness of hypothetical double-mutants. Based on these findings, the authors concluded that the product function is a robust tool for analyzing epistatic interactions in growth fitness and effectively reflects how growth rates depend on the combination of multiple biochemical pathways.

Strength

By leveraging a previously published large dataset of yeast colony sizes for single- and double-knockout mutants, this study validated the relevance of the product function, which has frequently been used in genetics to analyze epistatic interactions. The confirmation that the product function provides a more reliable prediction of double-mutant fitness compared to other neutral functions is valuable for researchers analyzing epistatic interactions, particularly those working with the same dataset.

Notably, this dataset has been previously used in studies exploring epistatic interactions with the product neutrality function. This study's findings affirm the validity of using the product function, which could enhance confidence in the conclusions drawn by those earlier studies. Consequently, both researchers utilizing this dataset and readers of prior research will benefit from the confirmation provided by this study.

Weakness

This study contains several serious problems, primarily stemming from the following issues: ignoring the substantial differences in the mechanisms regulating cell growth between prokaryotes and eukaryotes and adopting an overly specific and unrealistic set of assumptions in the mutation model. Below, the details are discussed.

(1) Misapplication of prokaryotic growth models

The mechanistic origin of the multiplicative model observed in yeast colony fitness is explained using a bacterial cell growth model. However, there is no valid justification for linking these two systems. The bacterial growth model, the Scott-Hwa model, heavily rely on specific molecular mechanisms, such as ppGpp-mediated regulation, which adjusts ribosome expression and activity during translation. In particular, this mechanism is critical to ensure growth-dependency of the fraction of ribosome in proteome in the Scott-Hwa model [https://doi.org/10.1111/j.1462-2920.2010.02357.x; https://doi.org/10.1073/pnas.2201585119]. Yeast cells lack this regulatory mechanism, making it inappropriate to directly apply bacterial growth models to yeast.

The Weiße model is based on a larger set of underlying equations and involves more parameters than the Scott-Hwa model. In the original paper by Weiße et al. (PNAS, 2015), however, the model parameters were fitted solely to experimental data from *E. coli*, and the model's applicability to yeast was never assessed. In summary, for neither the Scott-Hwa model nor the Weiße model has it been demonstrated that the entire model quantitatively fits experimental data from yeast. A positive correlation between growth rate and RNA/protein ratio, often observed in yeast, supports only a limited portion of either model, and does not constitute validation of the models as a whole.

(2) Overly specific assumptions in the theoretical model

The theoretical model assumes that two mutations affect only independent parameters of specific biochemical processes. However, this overly restrictive assumption weakens the model's validity in explaining the general occurrence of the multiplicative model in mutations. Furthermore, experimental evidence suggests limitations of this approach. For example, in most viable yeast deletion mutants with reduced growth rates, the expression of ribosomal proteins remained largely unchanged, contrary to the predictions of the Scott-Hwa model [https://doi.org/10.7554/eLife.28034]. This discrepancy highlights that the Scott-Hwa model and its derivatives cannot reliably explain mutants' growth rates based on current experimental evidence.

(3) Limited reliability of the mechanistic origin of the multiplicative model

The authors seem to regard growth-optimizing feedback as the mechanistic origin of the multiplicative model. However, the importance of growth-optimizing feedback in explaining product neutrality heavily depends on the very specific framework of the Scott-Hwa model. As I pointed out above, the Scott-Hwa model is a bacterial growth model that considers only a narrowly defined set of biochemical reactions. Using such a narrow model to explore the mechanistic origin of product neutrality observed on a genome-wide scale appears to be inappropriate. Arguments based on either the Scott-Hwa model or the Weiße model fail to account for the generality of product neutrality across diverse genetic perturbations. These models, in their current form, do not explain the broader patterns of product neutrality observed experimentally.

---

## [Referee Report · Reviewer #2 (Public review)]

The paper deals with the important question of gene epistasis, focusing on asking what is the correct null model for which we should declare no epistasis.

In the first part, they use the Synthetic Genetic Array dataset to claim that the effects of a double mutation on growth rate is well predicted by the product of the individual effects (much more than e.g. the additive model). The second (main) part shows this is also the prediction of two simple, coarse-grained models for cell growth.

I find the topic interesting, the paper well written, and the approach innovative.

Comments on revisions:

The authors have adequately addressed the comments raised in the review below, and I find that the paper has improved.

---

## [Author Response]

The following is the authors’ response to the original reviews

**Reviewer #1 (Public review):**
Summary:Detecting unexpected epistatic interactions among multiple mutations requires a robust null expectation - or neutral function - that predicts the combined effects of multiple mutations on phenotype, based on the effects of individual mutations. This study assessed the validity of the product neutrality function, where the fitness of double mutants is represented as the multiplicative combination of the fitness of single mutants, in the absence of epistatic interactions. The authors utilized a comprehensive dataset on fitness, specifically measuring yeast colony size, to analyze epistatic interactions.The study confirmed that the product function outperformed other neutral functions in predicting the fitness of double mutants, showing no bias between negative and positive epistatic interactions. Additionally, in the theoretical portion of the study, the authors applied a wellestablished theoretical model of bacterial cell growth to simulate the growth rates of both single and double mutants under various parameters. The simulations further demonstrated that the product function was superior to other functions in predicting the fitness of hypothetical double mutants. Based on these findings, the authors concluded that the product function is a robust tool for analyzing epistatic interactions in growth fitness and effectively reflects how growth rates depend on the combination of multiple biochemical pathways.Strengths:By leveraging a previously published extensive dataset of yeast colony sizes for single- and double-knockout mutants, this study validated the relevance of the product function, commonly used in genetics to analyze epistatic interactions. The finding that the product function provides a more reliable prediction of double-mutant fitness compared to other neutral functions offers significant value for researchers studying epistatic interactions, particularly those using the same dataset.Notably, this dataset has previously been employed in studies investigating epistatic interactions using the product neutrality function. The current study's findings affirm the validity of the product function, potentially enhancing confidence in the conclusions drawn from those earlier studies. Consequently, both researchers utilizing this dataset and readers of previous research will benefit from the confirmation provided by this study's results.Weaknesses:This study exhibits several significant logical flaws, primarily arising from the following issues: a failure to differentiate between distinct phenotypes, instead treating them as identical; an oversight of the substantial differences in the mechanisms regulating cell growth between prokaryotes and eukaryotes; and the adoption of an overly specific and unrealistic set of assumptions in the mutation model. Additionally, the study fails to clearly address its stated objective-investigating the mechanistic origin of the multiplicative model. Although it discusses conditions under which deviations occur, it falls short of achieving its primary goal. Moreover, the paper includes misleading descriptions and unsubstantiated reasoning, presented without proper citations, as if they were widely accepted facts. Readers should consider these issues when evaluating this paper. Further details are discussed below.(1) Misrepresentation of the dataset and phenotypesThe authors analyze a dataset on the fitness of yeast mutants, describing it as representative of the Malthusian parameter of an exponential growth model. However, they provide no evidence to support this claim. They assert that the growth of colony size in the dataset adheres to exponential growth kinetics; in contrast, it is known to exhibit linear growth over time, as indicated in [Supplementary Note 1 of https://doi.org/10.1038/nmeth.1534]. Consequently, fitness derived from colony size should be recognized as a different metric and phenotype from the Malthusian parameter. Equating these distinct phenotypes and fitness measures constitutes a fundamental error, which significantly compromises the theoretical discussions based on the Malthusian parameter in the study.

The reviewer is correct in pointing out that colony-size measurements are distinct from exponential growth kinetics. We acknowledge that our original text implied that the dataset directly measured the exponential growth rate (Malthusian parameter), when in fact it was measuring yeast colony expansion rates on solid media. Colony growth under these conditions often follows a biphasic pattern in that there is typically an initial microscopic phase where cells can grow exponentially, but as the colony expands further then the growth dynamics become more linear (Meunier and Choder 1999). We have revised our text to state clearly what the experiment measured.

However, while colony size does not exhibit exponential growth kinetics, several studies have argued that the rate of colony expansion is related to the exponential growth rate of cells growing in non-limiting nutrient conditions in liquid culture. This is because colony growth is dominated by cells at the colony boundaries that have access to nutrients and are in exponential growth. Cells in the colony interior lack nutrients and therefore contribute little to colony growth. This has been shown both in theoretical and experimental studies, finding that the linear growth rate of the colony is directly linked to the single-cell exponential growth rate (Pirt 1967; Gray and Kirwan 1974; Korolev et al. 2012; Gandhi et al. 2016; Meunier and Choder 1999). In particular, the above studies suggest that the linear colony growth rate is directly proportional to the square root of the exponential growth rate. Therefore, one would expect that the validity of the product model for one fitness measure implies its validity for the other measure. In addition, colony size was found to be highly correlated with the exponential growth rate of cells in non-limiting nutrients in liquid cultu

re (Baryshnikova et al. 2010; Zackrisson et al. 2016; Miller et al. 2022). For these reasons, we treated the colony size and exponential growth rate as interchangeable in our original manuscript.

However, while colony size does not exhibit exponential growth kinetics, several studies have argued that the rate of colony expansion is related to the exponential growth rate of cells growing in non-limiting nutrient conditions in liquid culture. This is because colony growth is dominated by cells at the colony boundaries that have access to nutrients and are in exponential growth. Cells in the colony interior lack nutrients and therefore contribute little to colony growth. This has been shown both in theoretical and experimental studies, finding that the linear growth rate of the colony is directly linked to the single-cell exponential growth rate (Pirt 1967; Gray and Kirwan 1974; Korolev et al. 2012; Gandhi et al. 2016; Meunier and Choder 1999). In particular, the above studies suggest that the linear colony growth rate is directly proportional to the square root of the exponential growth rate. Therefore, one would expect that the validity of the product model for one fitness measure implies its validity for the other measure. In addition, colony size was found to be highly correlated with the exponential growth rate of cells in non-limiting nutrients in liquid culture (Baryshnikova et al. 2010; Zackrisson et al. 2016; Miller et al. 2022). For these reasons, we treated the colony size and exponential growth rate as interchangeable in our original manuscript.

To address the important point raised by the reviewer, we now explain more clearly in the text what the analyzed data on colony size show and why we believe it is reflective of the exponential growth rate. Finally, we note that our results supporting the product neutrality function are consistent with the work of (Mani et al. 2008), which used smaller datasets based on liquid culture growth rates (Jasnos and Korona 2007; Onge et al. 2007).

The text in Section 2.3 now reads:

“Having verified empirically that the Product neutrality function is supported by the latest data for cell proliferation, we now turn our attention to its origins. Addressing this question requires some mechanistic model of biosynthesis. However, most mechanistic models of growth apply directly to single cells in rich nutrient conditions, which may not directly apply to the SGA measurements of colony expansion rates. In particular, colony growth has been shown to follow a biphasic pattern (Meunier et al. 1999). A first exponential phase is followed by a slower linear phase as the colony expands. Previous modeling and empirical work indicates that this second linear expansion rate reflects the underlying exponential growth of cells in the periphery of the colony (Pirt 1967; Gray et al. 1974; Gandhi et al. 2016; Baryshnikova, Costanzo, S. Dixon, et al. 2010; Zackrisson et al. 2016; Miller et al. 2022). More precisely, mathematical models show the linear colony-size expansion rate is directly proportional to the square root of the exponential growth rate under non-limiting conditions. Intuitively, this relationship arises because colony growth is dominated by the expansion of the population of cells in an annulus at the colony border that are exposed to rich nutrient conditions. These cells expand at a rate similar to the exponential rate of cells growing in a rich nutrient liquid culture. In contrast, the cells in the interior of the colony experience poor nutrient conditions, grow very slowly, and do not contribute to colony growth.

This intimate relationship between both proliferation rates allows us to explore the origin of the Product neutrality function in mechanistic models of cell growth. Indeed, if colony-based fitnesses follow a Product model, then\begin{document}$$\displaystyle W_{x y}^{c} \sim W_{x}^{c} W_{y}^{c} \Leftrightarrow \frac{\lambda_{x y}^{c}}{\lambda_{W T}^{c}} \sim \frac{\lambda_{x}^{c} \lambda_{y}^{c}}{\left(\lambda_{W T}^{c}\right)^{2}}$$\end{document}

where the superscript *c* indicates colony-based values for the fitness *W* and the growth rate *λ*. Taking into account the relationship between single-cell exponential growth rates and colony growth rates, we can write\begin{document}$$\displaystyle \lambda^{c} \propto \sqrt{\lambda^{l}}$$\end{document}

where the superscript l denotes liquid cultures. Combining these expressions, we obtain\begin{document}$$\displaystyle \frac{\sqrt{\lambda_{x y}^{l}}}{\sqrt{\lambda_{W T}^{l}}} \sim \frac{\sqrt{\lambda_{x}^{l}} \sqrt{\lambda_{y}^{l}}}{{\sqrt{\lambda_{W T}^{l}}}^{2}} \Rightarrow W_{x y}^{l} \sim W_{x}^{l} W_{y}^{l}$$\end{document}

In other words, from the perspective of the Product neutrality function, fitnesses based on colony expansion rates are equivalent to fitnesses based on single-cell exponential growth rates. The prevalence of the Product neutrality model—both in the SGA data and in previous studies on datasets from liquid cultures (Jasnos et al. 2007; Onge et al. 2007; Mani et al. 2008)—encourages the exploration of its origin in mechanistic models of cell growth.”

(2) Misapplication of prokaryotic growth modelsThe study attempts to explain the mechanistic origin of the multiplicative model observed in yeast colony fitness using a bacterial cell growth model, particularly the Scott-Hwa model. However, the application of this bacterial model to yeast systems lacks valid justification. The Scott-Hwa model is heavily dependent on specific molecular mechanisms such as ppGppmediated regulation, which plays a crucial role in adjusting ribosome expression and activity during translation. This mechanism is pivotal for ensuring the growth-dependency of the ribosome fraction in the proteome, as described in [https://doi.org/10.1073/pnas.2201585119]. Unlike bacteria, yeast cells do not possess this regulatory mechanism, rendering the direct application of bacterial growth models to yeast inappropriate and potentially misleading. This fundamental difference in regulatory mechanisms undermines the relevance and accuracy of using bacterial models to infer yeast colony growth dynamics.If the authors intend to apply a growth model with macroscopic variables to yeast double-mutant experimental data, they should avoid simply repurposing a bacterial growth model. Instead, they should develop and rigorously validate a yeast-specific growth model before incorporating it into their study.

There is nothing that is prokaryote specific in the Scott-Hwa model. It does not include the specific ppGpp mechanism to regulate ribosome fraction that does not exist in eukaryotes. The general features of the model, like how the ribosome fraction is proportional to the growth rate have indeed been validated in yeast (Metzl-Raz et al. 2017; Elsemman et al. 2022; Xia et al. 2022). Performing a detailed physiological analysis of budding yeast across varying growth conditions in order to build a more extensive model is beyond the scope of this work. Finally, we note that the Weiße model, which we also analyzed, is also generic and has replicated empirical measurements both from bacteria and yeast (Weiße et al. 2015).

To clarify this point in the text, we have added the following to Section 2.3:

“Experimental measurements in other organisms suggest that the observations leading to this model, including that the cellular ribosome fraction increases with growth rate, are in fact generic and also seen in the yeast *S. cerevisiae* (Metzl-Raz et al. 2017; Elsemman et al. 2022; Xia et al. 2022).”

(3) Overly specific assumptions in the theoretical modelhe theoretical model in question assumes that two mutations affect only independent parameters of specific biochemical processes, an overly restrictive premise that undermines its ability to broadly explain the occurrence of the multiplicative model in mutations. Additionally, experimental evidence highlights significant limitations to this approach. For example, in most viable yeast deletion mutants with reduced growth rates, the expression of ribosomal proteins remains largely unchanged, in direct contradiction to the predictions of the Scott-Hwa model, as indicated in [https://doi.org/10.7554/eLife.28034]. This discrepancy emphasizes that the ScottHwa model and its derivatives do not reliably explain the growth rates of mutants based on current experimental data, suggesting that these models may need to be reevaluated or alternative theories developed to more accurately reflect the complex dynamics of mutant growth.

In the data from the Barkai lab referenced by the reviewer (reproduced below), we see that the ribosomal transcript fraction is in fact proportional to growth rate in response to gene deletions in contradiction to the reviewer’s interpretation. However, it is notable that the ribosomal transcript fraction is a bit higher for a given growth rate if that growth rate is generated by a mutation rather than generated by a suboptimal nutrient condition. We know that the very simple Scott-Hwa model is not a perfect representation of the cell. Nevertheless, it does recapitulate important aspects of growth physiology and therefore we thought it is useful to analyze its response to mutations and compare those responses to the different neutrality functions. We never claimed the Scott-Hwa model was a perfect model and fully agree with the referee’s statement above that “... these models may need to be reevaluated, or alternative theories developed to more accurately reflect the complex dynamics of mutant growth.” Indeed, we say as much in our discussion where we wrote:

“While we focused on coarse-grained models for their simplicity and mechanistic interpretability, they might be too simple to effectively model large double-mutant datasets and the resulting double-mutant fitness distributions. We therefore expect the combination of high throughput genetic data with the analysis of larger-scale models, for instance based on Flux Balance Analysis, Metabolic Control Analysis, or whole-cell modeling, to lead to important complementary insights regarding the regulation of cell growth and proliferation.”

To further clarify this point, we discuss and cite the Barkai lab data for gene deletions see Figure 2 from Metzl-Raz et al. 2017.

(4) Lack of clarity on the mechanistic origin of the multiplicative modelThe study falls short of providing a definitive explanation for its primary objective: elucidating the "mechanistic origin" of the multiplicative model. Notably, even in the simplest case involving the Scott-Hwa model, the underlying mechanistic basis remains unexplained, leaving the central research question unresolved. Furthermore, the study does not clearly specify what types of data or models would be required to advance the understanding of the mechanistic origin of the multiplicative model. This omission limits the study's contribution to uncovering the biological principles underlying the observed fitness patterns.”

We appreciate the reviewer’s interest in a more complete mechanistic explanation for the product model of fitness. The primary goal of this study was to explore the validity of the Product model from the perspective of coarse-grained models of cell growth, and to extract mechanistic insights where possible. We view our work as a first step toward a deeper understanding of how double-mutant fitnesses combine, rather than a final, all-encompassing theory. As the referee notes, we are limited by the current state of the field, which has an incomplete understanding of cell growth.

Nonetheless, our analysis does propose concrete, mechanistically informed explanations. For example, we highlight how growth-optimizing feedback—such as cells’ ability to reallocate ribosomes or adjust proteome composition—naturally leads to multiplicative rather than additive or minimal fitness effects. We also link the empirical deviations from pure multiplicative behavior to differences in how specific pathways re-balance under perturbation, and we suggest that a product-like rule emerges when multiple interconnected processes each partially limit cell growth.

In the discussion, we clarify what additional data and models we think will be required to advance this question. Namely, we propose extending our approach through larger-scale, more detailed modeling frameworks – that may include explicit modeling of ppGpp or TOR activities in bacteria or eukaryotic cells, respectively. We also emphasize the importance of refining the measurement of cell growth rates to uncover subtle deviations from the product rule that could yield greater mechanistic insight. By integrating high-throughput genetic data with nextgeneration computational models, it should be possible to hone in on the specific biological principles (e.g., metabolic bottlenecks, resource reallocation) that underlie the multiplicative neutrality function.

**Reviewer #2 (Public review):**
The paper deals with the important question of gene epistasis, focusing on asking what is the correct null model for which we should declare no epistasis.In the first part, they use the Synthetic Genetic Array dataset to claim that the effects of a double mutation on growth rate are well predicted by the product of the individual effects (much more than e.g. the additive model). The second (main) part shows this is also the prediction of two simple, coarse-grained models for cell growth.I find the topic interesting, the paper well-written, and the approach innovative.One concern I have with the first part is that they claim that:"In these experiments, the colony area on the plate, a proxy for colony size, followed exponential growth kinetics. The fitness of a mutant strain was determined as the rate of exponential growth normalized to the rate in wild type cells."There are many works on "range expansions" showing that colonies expand at a constant velocity, the speed of which scales as the square root of the growth rate (these are called "Fisher waves", predicted in the 1940', and there are many experimental works on them, e.g. https://www.pnas.org/doi/epdf/10.1073/pnas.0710150104) If that's the case, the area of the colony should be proportional to growth_rate X time^2 , rather than exp(growth_rate*time), so the fitness they might be using here could be the log(growth_rate) rather than growth_rate itself? That could potentially have a big effect on the results.

We thank the reviewer for their thoughtful remarks. As they rightly pointed out, a large body of literature supports that colonies expand at constant velocity both from a theoretical and experimental standpoint.

As discussed in the answer to the first question of Reviewer 1, this body of work also suggests that the linear expansion rate of the colony front is directly related to the single-cell exponential growth rate of the cells at the periphery. Hence, although the macroscopic colony growth may not be exponential in time, measuring colony size (or radial expansion) across different genotypes still provides a consistent and meaningful proxy for comparing their underlying growth capabilities.

In particular, these studies suggest (consistently with Fisher-wave theory) that the linear growth rate of the colony 𝐾 is proportional to the square root of the exponential growth rate 𝜆. Under the assumption that the product model is valid for a given double mutant and for the exponential growth rate, we would have that\begin{document}$$\displaystyle W_{x y}=W_{x} W_{y} \Leftrightarrow \lambda_{x y}=\lambda_{x} \lambda_{y} / \lambda_{0}$$\end{document}

The associated wave-front velocities would then be predicted to be\begin{document}$$\displaystyle K_{x y} \propto \sqrt{ } \lambda_{x y} \Leftrightarrow K_{x y} \propto \sqrt{ } \lambda_{x} \sqrt{ } \lambda_{y} / \sqrt{ } \lambda_{0} \Leftrightarrow K_{x y} \propto K_{x} K_{y} / K_{0}$$\end{document}

In other words, if the product model is valid for fitness measures based on exponential growth rates, it should also be valid for fitness measures based on linear colony growth rates.

We now include this discussion in the revised version of Section 2.3.

Additional comments/questions:(1) What is the motivation for the model where the effect of two genes is the minimum of the two?

The motivation for the minimal model is the notion that there might be a particular process that is rate-limiting for growth due to a mutation. In this case, a mutation in process X makes it really slow and process Y proceeds in parallel and has plenty of time to finish its job before cell division takes place. In this case, even a mutation to process Y might not slow down growth because there is an excess amount of time for it to be completed. Thus, the double mutant might then be anticipated to have the growth rate associated with the single mutation to process X. We now add a similar description when we introduce the different neutrality functions in Section 2.1.

(2) How seriously should we take the Scott-Hwa model? Should we view it as a toy model to explain the phenomenon or more than that? If the latter, then since the number of categories in the GO analysis is much more than two (47?) in many cases the analysis of the experimental data would take pairs of genes that both affect one process in the Scott-Hwa model - and then the product prediction should presumably fail? The same comment applies to the other coarse-grained model.

From our perspective, models like the Scott-Hwa model constitute the simplest representation of growth based on data that is not trivial. Moreover, the Scott-Hwa model is able to incorporate interactions between two different biological processes. We believe models, like the Scott-Hwa and Weiße models, should be viewed as more than mere toy models because they have been backed up by some empirical data, such as that showing the ribosome fraction increases with growth rate. However, the Scott-Hwa model is inherently limited by its low dimensionality and relative simplicity. We do not claim that such models can provide a full picture of the cell. As argued in the main text, we have chosen to focus on such models because of their tractability and in the hope of extracting general principles. We nonetheless agree with the reviewer that they do not have the capacity to represent interactions between genes in the same biological process. We now note this limitation in the text.

(3) There are many works in the literature discussing additive fitness contributions, including Kaufmann's famous NK model as well as spin-glass-type models (e.g. Guo and Amir, Science Advances 2019, Reddy and Desai, eLife 2021, Boffi et al., eLife 2023) These should be addressed in this context.

We thank the reviewer for pointing out this part of the literature. We do believe these works constitute a relevant body of work tackling the emergence of epistasis patterns from a theoretical grounding, and now reference and discuss them in the text.

(4) The experimental data is for deletions, but it would be interesting to know the theoretical model's prediction for the expected effects of beneficial mutations and how they interact since that's relevant (as mentioned in the paper) for evolutionary experiments. Perhaps in this case the question of additive vs. multiplicative matters less since the fitness effects are much smaller.

This is an interesting question. Since mutations increasing the growth rate generated by gene deletions or other systematic perturbations are rare, we did not focus on them. Of course, as the reviewer notes, in the case of evolution experiments, these fitness enhancing mutations are selected for. To address the reviewer's question, we can first consider the Scott-Hwa model. In this case, the analytical solution remains valid in the case of fitness enhancing mutations so that the fitness of the double mutant will be the product neutrality function multiplied by an additional interaction term (see Figure 3). The mathematical derivation predicts that the double mutant fitness can potentially grow indefinitely. Indeed, the denominator can be equal to zero in some cases. In simulations, we see that the observation for deleterious mutations does not seem to hold for beneficial mutations (new supplementary Figure S5 shown below). Indeed, no model seems to replicate double mutant fitnesses much better than any other. This suggests that the growth-optimizing feedback we discuss in section 2.3 may have compound effects that ultimately make double-mutant fitnesses much larger than any model predicts.

We recognize this may be an important point, and discuss it in detail in the revised section 2.3 as well as in the discussion.

Baryshnikova, Anastasia, Michael Costanzo, Scott Dixon, Franco J. Vizeacoumar, Chad L. Myers, Brenda Andrews, and Charles Boone. 2010. “Synthetic Genetic Array (SGA) Analysis in *Saccharomyces cerevisiae* and *Schizosaccharomyces pombe*.” Methods in Enzymology 470 (March):145–79.

Elsemman, Ibrahim E., Angelica Rodriguez Prado, Pranas Grigaitis, Manuel Garcia Albornoz, ictoria Harman, Stephen W. Holman, Johan van Heerden, et al. 2022. “Whole-Cell Modeling in Yeast Predicts Compartment-Specific Proteome Constraints That Drive Metabolic Strategies.” Nature Communications 13 (1): 801.

Gandhi, Saurabh R., Eugene Anatoly Yurtsev, Kirill S. Korolev, and Jeff Gore. 2016. “Range Expansions Transition from Pulled to Pushed Waves as Growth Becomes More Cooperative in an Experimental Microbial Population.” Proceedings of the National Academy of Sciences of the United States of America 113 (25): 6922–27.

Gray, B. F., and N. A. Kirwan. 1974. “Growth Rates of Yeast Colonies on Solid Media.” Biophysical Chemistry 1 (3): 204–13.

Jasnos, Lukasz, and Ryszard Korona. 2007. “Epistatic Buffering of Fitness Loss in Yeast Double Deletion Strains.” Nature Genetics 39 (4): 550–54.

Korolev, Kirill S., Melanie J. I. Müller, Nilay Karahan, Andrew W. Murray, Oskar Hallatschek, and David R. Nelson. 2012. “Selective Sweeps in Growing Microbial Colonies.” Physical Biology 9 (2): 026008.

Mani, Ramamurthy, Robert P. St Onge, John L. Hartman 4th, Guri Giaever, and Frederick P. Roth. 2008. “Defining Genetic Interaction.” Proceedings of the National Academy of Sciences of the United States of America 105 (9): 3461–66.

Metzl-Raz, Eyal, Moshe Kafri, Gilad Yaakov, Ilya Soifer, Yonat Gurvich, and Naama Barkai. 2017. “Principles of Cellular Resource Allocation Revealed by Condition-Dependent Proteome Profiling.” eLife 6 (August). https://doi.org/10.7554/elife.28034.

Meunier, J. R., and M. Choder. 1999. “*Saccharomyces cerevisiae* Colony Growth and Ageing: Biphasic Growth Accompanied by Changes in Gene Expression.” Yeast (Chichester, England) 15 (12): 1159–69.

Miller, James H., Vincent J. Fasanello, Ping Liu, Emery R. Longan, Carlos A. Botero, and Justin C. Fay. 2022. “Using Colony Size to Measure Fitness in *Saccharomyces cerevisiae*.” PloS e 17 (10): e0271709.

Onge, Robert P. St, Ramamurthy Mani, Julia Oh, Michael Proctor, Eula Fung, Ronald W. Davis, Corey Nislow, Frederick P. Roth, and Guri Giaever. 2007. “Systematic Pathway Analysis Using High-Resolution Fitness Profiling of Combinatorial Gene Deletions.” Nature Genetics 39 (2): 199–206.

Pirt, S. J. 1967. “A Kinetic Study of the Mode of Growth of Surface Colonies of Bacteria and Fungi.” Journal of General Microbiology 47 (2): 181–97.

Weiße, Andrea Y., Diego A. Oyarzún, Vincent Danos, and Peter S. Swain. 2015. “Mechanistic Links between Cellular Trade-Offs, Gene Expression, and Growth.” Proceedings of the National Academy of Sciences of the United States of America 112 (9): E1038–47.

Xia, Jianye, Benjamin J. Sánchez, Yu Chen, Kate Campbell, Sergo Kasvandik, and Jens Nielsen. 2022. “Proteome Allocations Change Linearly with the Specific Growth Rate of *Saccharomyces cerevisiae* under Glucose Limitation.” Nature Communications 13 (1): 2819.

Zackrisson, Martin, Johan Hallin, Lars-Göran Ottosson, Peter Dahl, Esteban Fernandez-Parada, Erik Ländström, Luciano Fernandez-Ricaud, et al. 2016. “Scan-O-Matic: High-Resolution Microbial Phenomics at a Massive Scale.” G3 (Bethesda, Md.) 6 (9): 3003–14.